# The transcription factor GATA4 promotes myocardial regeneration in neonatal mice

Mona Malek Mohammadi[1], Badder Kattih[1], Andrea Grund[1], Natali Froese[1], Mortimer Korf-Klingebiel[1], Anna Gigina[1], Ulrike Schrameck[1], Carsten Rudat[2], Qiangrong Liang[3], Andreas Kispert[2,4], Kai C Wollert[1,4], Johann Bauersachs[1,4] & Joerg Heineke[1,4,*] [ID]

## Abstract

Heart failure is often the consequence of insufficient cardiac regeneration. Neonatal mice retain a certain capability of myocardial regeneration until postnatal day (P)7, although the underlying transcriptional mechanisms remain largely unknown. We demonstrate here that cardiac abundance of the transcription factor GATA4 was high at P1, but became strongly reduced at P7 in parallel with loss of regenerative capacity. Reconstitution of cardiac GATA4 levels by adenoviral gene transfer markedly improved cardiac regeneration after cryoinjury at P7. In contrast, the myocardial scar was larger in cardiomyocyte-specific *Gata4* knockout (CM-G4-KO) mice after cryoinjury at P0, indicative of impaired regeneration, which was accompanied by reduced cardiomyocyte proliferation and reduced myocardial angiogenesis in CM-G4-KO mice. Cardiomyocyte proliferation was also diminished in cardiac explants from CM-G4-KO mice and in isolated cardiomyocytes with reduced GATA4 expression. Mechanistically, decreased GATA4 levels caused the downregulation of several pro-regenerative genes (among them interleukin-13, *Il13*) in the myocardium. Interestingly, systemic administration of IL-13 rescued defective heart regeneration in CM-G4-KO mice and could be evaluated as therapeutic strategy in the future.

**Keywords** cardiac regeneration; cardiomyocyte proliferation; GATA4; IL-13; neonatal cryoinfarction

**Subject Categories** Cardiovascular System; Regenerative Medicine; Stem Cells

## Introduction

The prevalence of chronic heart failure (CHF) as long-term sequel of myocardial infarction (MI) is steadily rising (Velagaleti *et al*, 2008).

Similarly, pediatric patients with congenital heart disease even after surgical correction are often at risk to develop CHF (Bolger *et al*, 2003). This is mainly due to the low endogenous regenerative capacity of the mammalian heart and the failure to replace lost myocardium with new contractile cardiomyocytes. In fact, cardiomyocyte renewal was demonstrated to occur only at a rate of 0.5–1% per year in adult humans (Bergmann *et al*, 2009). Consequently, a collagen rich scar forms after injury in the heart, which ultimately favors ventricular dilatation and failure (Fraccarolo *et al*, 2012). While current treatment approaches mainly target these cardiac remodeling processes, the promotion of endogenous regenerative mechanisms could also be considered as treatment strategy. Although still somewhat controversial, newborn mice are able to almost completely regenerate their myocardium after apex resection or myocardial infarction, while only partial regeneration was found after cryoinjury (Porrello *et al*, 2011, 2013; Haubner *et al*, 2012; Jesty *et al*, 2012; Mahmoud *et al*, 2013; Xin *et al*, 2013; Andersen *et al*, 2014; Porrello & Olson, 2014; Konfino *et al*, 2015; Leone *et al*, 2015; Polizzotti *et al*, 2015, 2016). Similar to the situation in zebrafish, which are also capable of unmitigated myocardial regeneration, restoration of the myocardium in neonatal mice is mainly achieved by proliferation of healthy cardiomyocytes from the edge of the injured area (Jopling *et al*, 2010; Porrello *et al*, 2011; Senyo *et al*, 2013). In addition, macrophage influx and angiogenesis are necessary to enable efficient myocardial regeneration in zebrafish and mice (Lepilina *et al*, 2006; Huang *et al*, 2013; Aurora *et al*, 2014). A case of a newborn child with severe myocardial infarction directly after birth, but complete cardiac recovery and normalization of heart function within weeks, suggested that neonatal cardiac regeneration might even be possible in humans (Haubner *et al*, 2016). The cardiac regenerative capacity, however, becomes strongly diminished at postnatal day (P)7 in mice (Porrello *et al*, 2011). Molecular mechanisms of mammalian neonatal regeneration are only beginning to become deciphered (Porrello *et al*, 2011, 2013; Jesty *et al*, 2012; Mahmoud *et al*, 2013, 2015; Xin *et al*, 2013; Aurora *et al*, 2014; Porrello & Olson, 2014; D'Uva *et al*, 2015; Polizzotti *et al*, 2015), and above all, it remains elusive, why the

1 Klinik für Kardiologie und Angiologie, Medizinische Hochschule Hannover, Hannover, Germany
2 Institut für Molekularbiologie, Medizinische Hochschule Hannover, Hannover, Germany
3 Department of Biomedical Sciences, New York Institute of Technology College of Osteopathic Medicine, Old Westbury, NY, USA
4 Cluster of Excellence REBIRTH, Medizinische Hochschule Hannover, Hannover, Germany
*Corresponding author. Tel: +49 511 532 3079; Fax: +49 511 532 5412; E-mail: heineke.joerg@mh-hannover.de

regenerative capacity is lost around P7. Identification of transcriptional regulators that promote cardiac reconstitution after injury directly after birth, but become downregulated around P7, could contribute to the development of novel therapeutic strategies aiming to improve regeneration and thereby reduce cardiac scarring in patients.

GATA4 belongs to the GATA family of transcription factors, of which six GATA factors exist in vertebrates, all sharing a conserved two-zinc finger-containing DNA binding domain and all binding to the motif (A/T)GATA(A/G) (Molkentin, 2000). Cardiomyocyte GATA4 plays an essential role to promote intrauterine cardiac development, for example, by driving the proliferation of fetal cardiac myocytes (Zeisberg *et al*, 2005; Rojas *et al*, 2008; Singh *et al*, 2010; Trivedi *et al*, 2010). In this regard, very early embryonic deletion of cardiomyocyte GATA4 in *Nkx2.5-Cre/+; Gata4^{flox/flox}* mice (with complete elimination of GATA4 by embryonic day E9.5) led to embryonic lethality with myocardial hypoplasia due to reduced cardiomyocyte proliferation (Zeisberg *et al*, 2005). In contrast, later elimination of cardiomyocyte GATA4 by the time fetal myocyte proliferation is winding down at around E18 (with the use of *Tg(β-MHC-Cre);Gata4^{flox/flox}* mice) did not result in cardiac hypoplasia, embryonic lethality, or any cardiac phenotype until the age of 12 weeks, when heart failure starts to develop (Oka *et al*, 2006). In the adult mouse heart, GATA4 plays a major role to promote cardiac hypertrophy and cardiac angiogenesis and to maintain cardiac function during pathological pressure overload, which also leads to increased cardiac GATA4 protein abundance (Bisping *et al*, 2006; Oka *et al*, 2006; Heineke *et al*, 2007). Recently, a prominent role of cardiomyocyte GATA4 was suggested for myocardial regeneration in zebrafish, because GATA4 was strongly induced selectively in proliferating cardiomyocytes that repopulated the cardiac apex after resection and inhibition of GATA4 blunted myocardial regeneration in adult zebrafish (Kikuchi *et al*, 2010; Gupta *et al*, 2013).

Here, we examined the impact of cardiomyocyte GATA4 for mammalian neonatal heart regeneration. We found that cardiac GATA4 protein is abundant in mice shortly after birth, but becomes dramatically reduced at P7, when also the regenerative capacity of the heart is diminished. Cardiomyocyte-specific genetic deletion of *Gata4* impedes cardiac regeneration after myocardial cryoinjury at P0 and, in turn, replenishment of GATA4 levels at P7 led to improved cardiac regeneration at this later stage.

# Results

## Cardiomyocyte GATA4 is necessary for neonatal heart regeneration

The analysis of cardiac GATA4 abundance by immunoblotting revealed high myocardial GATA4 levels on P1, which were strongly reduced by P7 and remained low with mild further reduction until P60 (Fig 1A and B). Cardiac GATA4 protein levels were also partially diminished 2 days after cryoinjury (Fig EV1A and B). Since the postnatal reduction of GATA4 levels at P7 paralleled the reported loss of myocardial regenerative capacity, we analyzed the functional relevance of high endogenous cardiac GATA4 in neonatal mice for heart regeneration by using cardiomyocyte-specific *Gata4* knockout mice (*Tg(β-MHC-Cre);Gata4^{flox/flox}*, short: CM-G4-KO). As shown by immunoblotting, CM-G4-KO mice displayed strongly reduced cardiac GATA4 protein abundance at P1 compared to control mice (Fig 1C and D). Immunofluorescence analysis revealed that GATA4 was deleted specifically in cardiomyocytes in CM-G4-KO mice at P1 (Fig 1E). A left ventricular cryoinfarction was induced at the day of birth (P0) in control and CM-G4-KO mice with a standardized cryoprobe. Interestingly, CM-G4-KO mice exerted significantly larger myocardial scars compared to control mice at 7, 21, and 60 days after cryoinfarction, suggesting that cardiomyocyte GATA4 is necessary for efficient myocardial regeneration (Fig 1F–H). Echocardiography at day 7 after injury revealed reduced cardiac function in CM-G4-KO mice compared to WT mice, but no differences between both genotypes after sham surgery (Fig 1I). Importantly, no difference in cardiac scar size after cryoinjury was noted between wild-type mice with and without β-MHC-Cre, excluding an influence of cardiomyocyte Cre-recombinase expression on scar formation (Fig EV1C). In addition, administration of GATA4 encoding adenovirus (Ad.GATA4 versus control adenovirus Ad.Control) to the myocardium of CM-G4-KO mice directly after the induction of cryoinjury at P0 resulted in replenishment of cardiac GATA4 level at P7 toward the high levels observed in control mice at P1 (Fig EV1D and E) and thereby significantly reduced the cardiac scar size in CM-G4-KO mice (Fig EV1F). These data imply that impaired myocardial regeneration in CM-G4-KO mice is directly due to reduced GATA4 abundance and not an indirect consequence of complications during development after E18. Although the remaining cardiac scar 60 days after cryoinjury is still larger in

---

**Figure 1.  Cardiac GATA4 becomes downregulated within the first postnatal week and is necessary for myocardial regeneration.**

A   Cardiac GATA4 protein abundance analyzed by immunoblotting. GAPDH was the loading control. + denotes positive control for GATA4 from cardiomyocytes infected with a GATA4-overexpressing adenovirus.

B   Densitometric quantification of the immunoblot shown in (A); ***$P$ = 0.0003 for P1 vs. P7 and P1 vs. P21; ***$P$ < 0.0001 for P1 vs. P60.

C   Immunoblot of cardiac GATA4 abundance in control (Con) and cardiomyocyte-specific *Gata4* knockout mice (CM-G4-KO) at postnatal day (P)1.

D   Quantification of the immunoblot shown in (C); *$P$ = 0.018.

E   Cardiac immunofluorescence staining and its quantification from Con and CM-G4-KO mice at P1. Scale bars: 25 μm; ****$P$ < 0.0001.

F   Transversal sections of mouse hearts 7 days after cryoinjury as indicated, stained with Sirius red. Scale bars: 500 μm. The quantification of the left ventricular (LV) scar area (red, arrows) 7 days after injury is shown on the right; **$P$ = 0.0027.

G   The quantification of the LV scar area of the indicated mice 21 days after cryoinjury; *$P$ = 0.0261.

H   The quantification of the LV scar area of the indicated mice 60 days after cryoinjury; ****$P$ < 0.0001.

I   Echocardiographic analysis of left ventricular systolic function (measured as fractional area change) in the indicated mice 7 days after cryoinjury; *$P$ = 0.033 between Con mice after sham or cryoinjury and *$P$ = 0.04 between Con and CM-G4-KO mice after cryoinjury.

Data information: (B, D–I) The number within bars indicates the number of mice analyzed in that particular group. All data are expressed as mean ± SEM. Unpaired Student's *t*-test (D–H) and one-way ANOVA with Sidak's multiple comparisons test (B, I) were used to compare groups.
Source data are available online for this figure.

CM-G4-KO mice compared to control mice (Fig 1H), this does not result in any measurable difference in systolic heart function at this adult stage, perhaps as a result of partial redundancy in myocardial regenerative mechanisms, which still promote a considerable degree of cardiac reconstitution toward adulthood even in the absence of GATA4 (Fig EV1G).

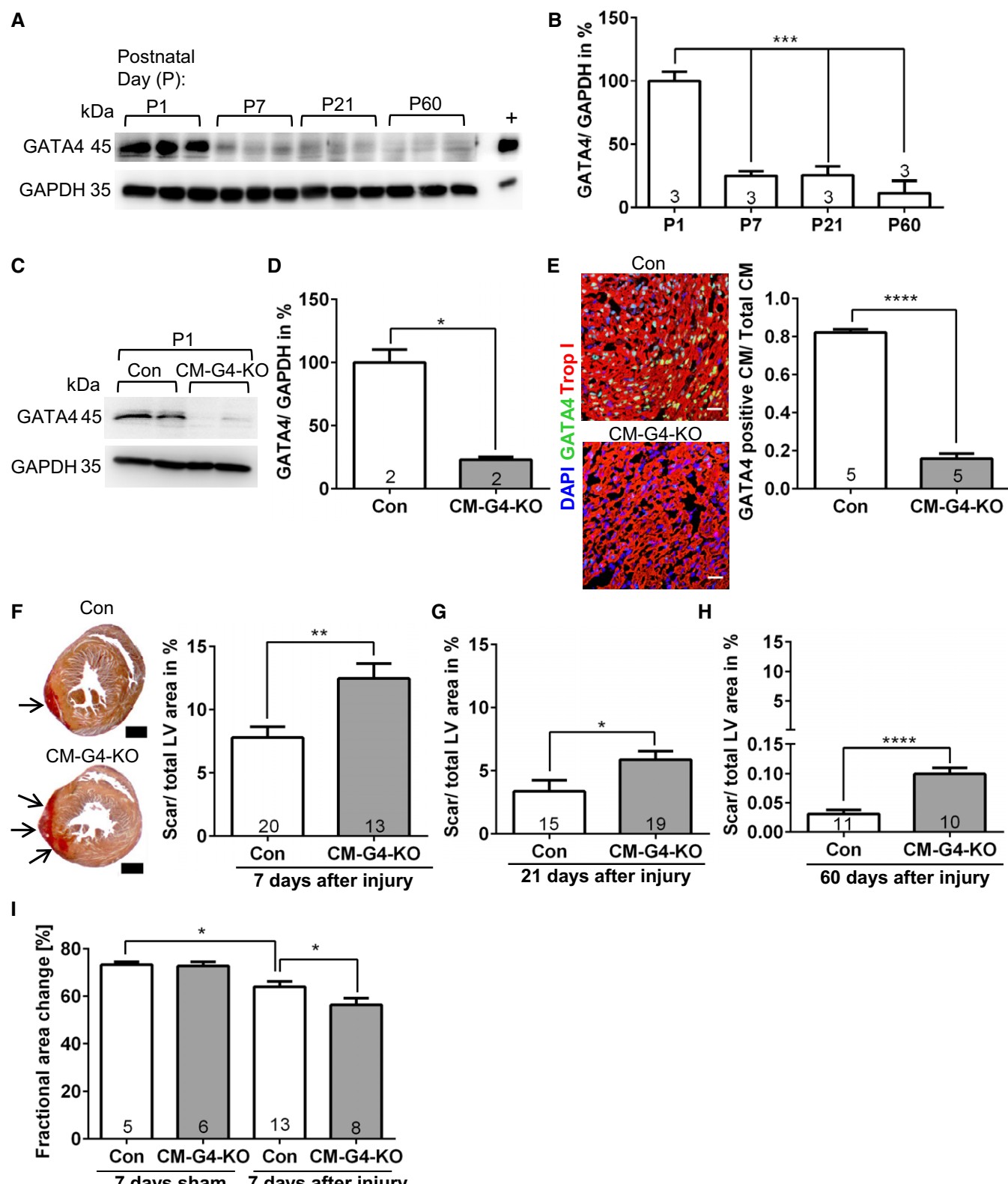

**Figure 1.**

## Loss of *Gata4* entails reduced angiogenesis and cardiomyocyte proliferation after cryoinfarction

Next, we wanted to identify the mechanisms underlying hampered myocardial regeneration in the *Gata4* mutant mice. We verified by TUNEL staining that the initial cardiac injury size was indeed not different between CM-G4-KO and control mice 3 h after cryoinjury (Fig 2A). We found a reduced abundance of CD31-positive myocardial capillaries outside the injury zone of CM-G4-KO compared to control mice 7 days after injury, but not after sham surgery (Fig 2B). These results were independently confirmed by quantitative real-time PCR, which revealed a significantly reduced *Cd31* mRNA expression in the myocardium of CM-G4-KO mice after cryoinjury (Fig EV2A). Immunofluorescence staining for the mitosis marker phospho-histone H3 (pH3) and co-staining for the cardiomyocyte-specific protein troponin T demonstrated a decreased rate of cardiomyocyte mitosis in CM-G4-KO versus control mice 7 days after infarction in the surviving myocardium (Fig 2C). We analyzed aurora B-stained cardiac tissue sections to detect cardiomyocyte cell divisions and found reduced cardiomyocyte cytokinesis in CM-G4-KO mice 7 days after cryoinjury (Fig 2D). Interestingly, cardiomyocyte mitosis and cell division were already strongly reduced 1 day after cryoinjury (Fig EV2B and C). In contrast, we found similar numbers of macrophages (F4/80 positive) within the injury and the remote area in CM-G4-KO and control mice (Fig EV2D). We also found no differences in cardiac hypertrophy between control and CM-G4-KO mice 3 h and 7 days after cryoinjury or sham surgery when analyzing the heart weight/body weight ratio (HW/BW), the cardiomyocyte cross-sectional area, or hypertrophy-related gene expression (*Nppa*, *Nppb*; Fig EV2E–H). Probing of cardiac cell type-specific gene expression by qPCR from heart tissue excluded major differences in the abundance of c-kit-positive resident cardiac stem cells, monocytes/macrophages (*Cd14*), CD4 T cells, or epicardial cells (*Wt1, Raldh2, Tcf21*) between control and CM-G4-KO mice after cryoinfarction (Fig EV3A). CD86 as marker of pro-inflammatory (M1-type) macrophages was not differentially regulated between the groups, while CD206 (marker of the tissue repair-promoting M2-macrophages) was similarly upregulated in the hearts of control and CM-G4-KO mice after cryoinjury. With regard to cardiomyocyte myosin heavy chain (MHC) gene expression, no change in *Myh6* (α-MHC) but a significant increase in *Myh7* (β-MHC) mRNA was noted in CM-G4-KO mice, which was

previously reported (Oka *et al*, 2006). Furthermore, epicardial activation was assessed by *in situ* hybridization for *Tbx18*, *Tcf21,* and *Raldh2* mRNA, but no epicardial activation was seen in control or CM-G4-KO mice after injury (Fig EV3B). Immunofluorescence staining failed to identify c-kit-positive cells in the myocardium of control or *Gata4* mutant mice (Fig EV3C). In support of the qPCR and *in situ* hybridization data, no differences in epicardial thickness were noticeable after wt-1 immunostaining between these groups (Fig EV3D), which also did not show any CD3-positive T cells in the myocardium (Fig EV3E).

## Disturbed regenerative gene expression in *Gata4* mutant mice

Since therefore decreased cardiomyocyte proliferation and angiogenesis are the most likely reasons for impaired heart regeneration in CM-G4-KO mice, we employed quantitative real-time PCR (qPCR) to profile the myocardial expression of candidate genes known to be involved in these processes (Fig 2E and Appendix Table S1). Several cell cycle and cell division-promoting genes were downregulated after sham surgery (*Ccna2*, *Ccne1*, *Cdk4*, *Cenpa*, Cdc2, *E2f1*) and/or after cryoinjury (*Ccna2*, *Zfp191*, *Cenpa*) in the myocardium of CM-G4-KO mice. In addition, some cell cycle-inhibiting genes were either upregulated (*Tsc22d1*) or downregulated (*Rbl1*, *Rab3gap1*, and *Meis1*) after cryoinfarction in CM-G4-KO mice. With regard to the expression of cytokines or receptors putatively promoting regeneration or angiogenesis, we found a significantly reduced expression of *Vegfa*, *Il13,* and the Igf2 receptor (*Igf2r*) in these mice. In contrast, *Ctf1*, *Igf1*, *Fgf16,* and the anti-regenerative cytokine *Tgfb* were not significantly changed in their expression between the different conditions (Appendix Table S1). Overall, we observed a cardiac gene-expression pattern in CM-G4-KO mice that could explain reduced regeneration.

## GATA4 deficiency causes reduced proliferation in cardiomyocytes and cardiac explants

To directly assess the impact of GATA4 deficiency on cardiomyocyte proliferation, we downregulated GATA4 by adenoviral expression of a shRNA (Ad.shGATA4) in fetal rat cardiomyocytes at E17 (Fig 3A), which usually exert a high proliferative capacity. Treatment with Ad.shGATA4 blunted the increase in cardiomyocyte number over 48 h in a pre-selected low-magnification microscopic field in the

---

**Figure 2. Reduced angiogenesis and cardiomyocyte proliferation in CM-G4-KO mice.**

A Representative pictures of TUNEL, WGA (wheat germ agglutinin to mark cardiomyocyte membranes), and DAPI immunofluorescence staining of mouse hearts as indicated. The white dashed line encircles the infarcted area (arrows). Scale bars: 200 μm. The quantification of the area of TUNEL-positive cells 3 h after cryoinjury is shown on the right (A′).

B Representative merged and single-channel pictures of cardiac CD31, DAPI, and WGA immunostaining in mouse hearts as indicated. The quantification of the myocardial capillary density as capillaries/cardiomyocyte (CM) ratio 7 days (d) after cryoinjury or sham surgery is shown as bar graph (B′). Scale bars: 20 μm. **P = 0.0039.

C Representative pictures of myocardial immunofluorescence staining for the mitosis marker phospho-histone H3 (pH3), troponin T, and DAPI in mice as indicated. Enlarged pH3-positive cardiomyocyte nuclei are shown separately. The arrows indicate pH3-positive cardiomyocyte nuclei. Scale bars: 25 μm. The quantification of pH3-positive (+) cardiomyocytes per microscopic field is shown as bar graph (C′); *P = 0.046.

D Representative myocardial immunostaining for aurora B, DAPI, and troponin T (Trop T). The arrow points toward a cardiomyocyte cell division, in which aurora B localizes at the midbody in cytokinesis. Scale bar: 25 μm. The quantification of dividing cardiomyocytes per microscopic field is shown as bar graph (D′); *P = 0.0104.

E Determination of cardiac gene expression by qPCR 7 days after sham or cryoinjury for the indicated genes; *P = 0.039 for *Il13* and *P = 0.049 for *Vegfa*; **P = 0.0043 for *Ccne1*, **P = 0.0027 for *Cdk4*, **P = 0.0096 for *Cenpa*, **P = 0.0015 for *Zfp191*; ***P = 0.0009 for *Ccna2*, ***P = 0.0002 for *Cenpa*.

Data information: The number within bars indicates the number of mice analyzed in that particular group. All data are expressed as mean ± SEM. Unpaired Student's *t*-test (A′) and one-way ANOVA with Sidak's multiple comparisons test (B–E) were used to compare groups.

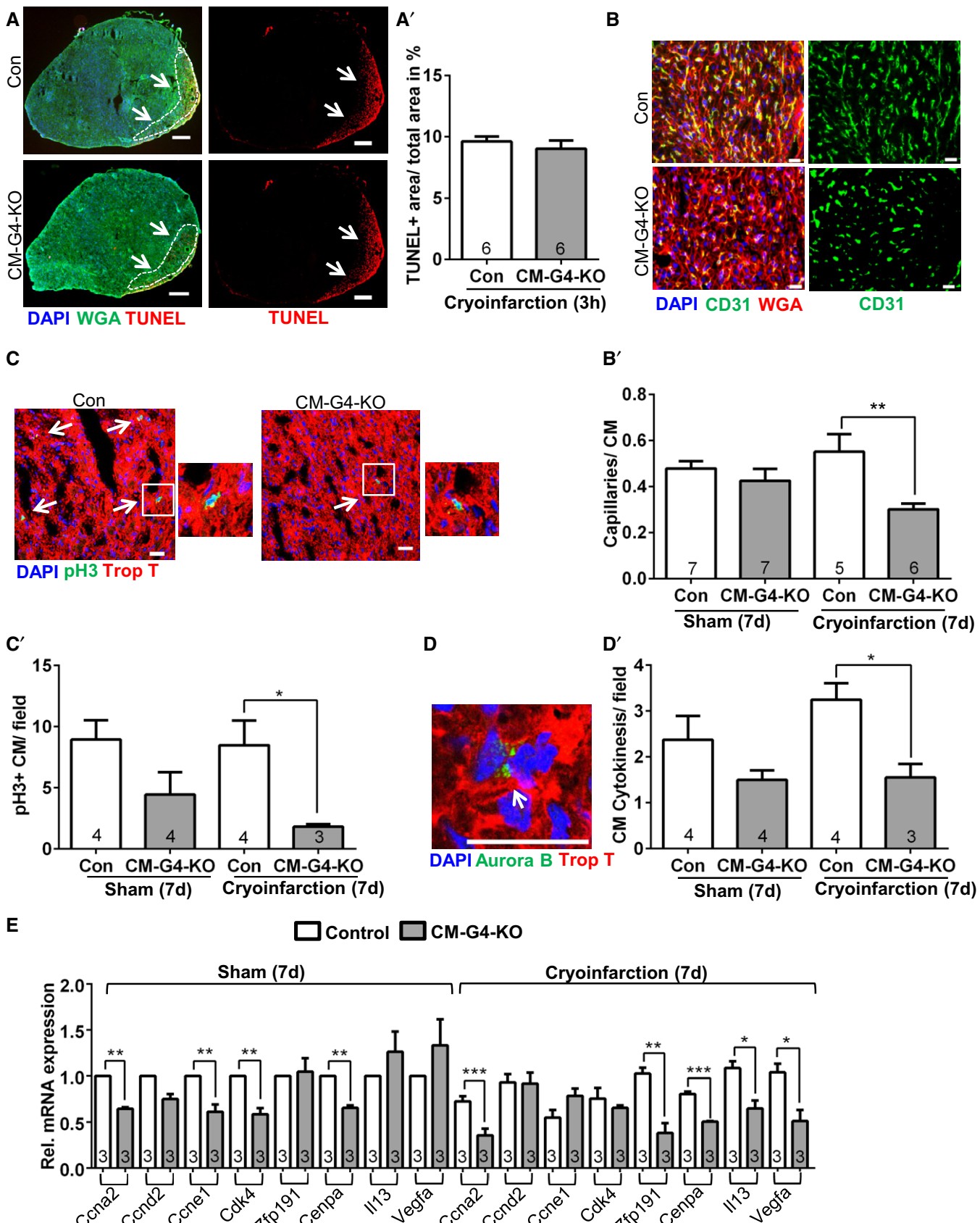

**Figure 2.**

presence of growth medium as compared to Ad.shControl-treated cells. This difference was likely the consequence of reduced cardiomyocyte proliferation, which was indicated by reduced DNA synthesis (BrdU incorporation, Fig 3C), reduced cardiomyocyte mitosis (Fig 3D), and a trend toward reduced cytokinesis in Ad.shGATA4-treated cells (Fig 3E). In contrast, cardiomyocyte apoptosis was not different between both conditions, as indicated by similar levels of cells staining positive for cleaved caspase 3 (Fig 3F). In addition, we used cardiac explant culture to assess cardiomyocyte outgrowth by embedding myocardial tissue pieces of control and CM-G4-KO mice 3 h after cryoinjury into Matrigel matrix. A markedly reduced number of troponin I-positive cardiomyocytes grew out of the myocardial tissue from CM-G4-KO mice, and less cardiomyocytes mitosis was observed compared to control tissue (Fig 3G and H). Because also cell migration plays a role in this assay, it is likely that reduced cardiomyocyte migration and reduced mitosis in combination led to diminished cardiomyocyte outgrowth from CM-G4-KO tissue.

## GATA4 overexpression promotes cardiomyocyte proliferation and myocardial regeneration

Since lack of GATA4 markedly impaired cardiac regeneration in mice, we aimed to analyze whether overexpression of GATA4 could improve cardiac regeneration. First, we assessed whether enhanced GATA4 expression by an adenoviral vector (Ad.GATA4, in comparison with control vector Ad.Control; Fig 4A) *in vitro* influenced the proliferation of isolated neonatal rat cardiomyocytes. As shown in Fig 4B, Ad.GATA4 significantly increased cardiomyocyte DNA synthesis, mitosis, and cell division (Fig 4B). This was accompanied (Appendix Table S2) by enhanced expression of cell cycle-promoting genes (*Ccne1*, *Cdk4*, *Cdc2*), reduced expression of the tumor suppressor gene *Tsc22d1*, and an increased expression of putatively pro-regenerative cytokines or receptors (*Il13*, *Ctf1*, *Igf2r*). Paradoxically, the gene encoding the cell cycle-inhibiting protein Rab3gap1 was upregulated, while the regenerative growth factor gene *Fgf16* was markedly suppressed as consequence of GATA4 overexpression (Appendix Table S2). To assess whether restoration of myocardial GATA4 abundance toward neonatal levels (i.e., P1)

could improve heart regeneration after P7, we injected Ad.GATA4 into the myocardium of wild-type mice at P7 directly after the induction of cryoinjury. This indeed led to an increased cardiac GATA4 abundance at P14, similar to what is usually present in the heart at P1 (Fig 4C and D). Immunofluorescence analysis revealed rather homogenous GATA4 expression in the heart 7 days after myocardial application of Ad.GATA4 (Fig 4E).

Interestingly, the increased GATA4 levels resulted in a significantly reduced myocardial scar size 7 days after cryoinfarction at P14 and was accompanied by increased cardiomyocyte cell cycle (Ki67 labeling) and mitotic activity, increased cardiomyocyte cytokinesis, and an increased capillary density (Fig 4F–I). In contrast, cardiac hypertrophy, pulmonary congestion, the abundance of macrophages, and α-smooth-muscle actin (αSMA)-positive small conductance vessels were not significantly changed by Ad.GATA4 treatment (Fig EV4A–F).

## Systemic application of IL-13 rescues the regenerative defects in *Gata4*-deficient mice

While the regenerative defects in CM-G4-KO mice after cryoinjury are likely the result of the combined changes in gene expression triggered directly or indirectly through the loss of *Gata4* in cardiomyocytes, we still aimed to elucidate whether compensation of reduced cardiac *Il13* mRNA expression in CM-G4-KO mice by exogenous, systemic administration of IL-13 could improve heart regeneration, since protein therapy with recombinant IL-13 might be feasible as therapeutic approach in the future. IL-13 was also recently identified as nodal upstream inducer of cardiomyocyte proliferation through a comprehensive gene-expression screen (O'Meara *et al*, 2015), and in addition, we found by ChIP assay that GATA4 directly interacts with the *Il13* promoter in the myocardium of neonatal wild-type mice after cryoinjury, therefore suggesting direct regulation of *Il13* expression by GATA4 (Fig EV4G). We therefore injected recombinant IL-13 intraperitoneally (or PBS as control) once daily four times from day 3 until day 6 after cryoinjury (Fig 5A). Intriguingly, IL-13 treatment strongly reduced cryoinjury-triggered cardiac scar formation and systolic dysfunction in CM-G4-KO mice and abolished the difference in scar size and cardiac function

---

**Figure 3.  Reduced GATA4 levels entail diminished cardiomyocyte proliferation *in vitro*.**

A   Representative immunoblots for GATA4 and GAPDH (as loading control) of fetal cardiomyocytes treated with adenoviruses (Ad) expressing control shRNA (shCon) or shRNA against *Gata4* (shGATA4) for 48 h.

B   Number of cardiomyocytes (CM) in a defined low-magnification (50×) microscopic field before the adenoviral infection and 48 h later. ****$P < 0.0001$.

C   Representative pictures and quantification of BrdU incorporation in fetal cardiomyocytes 48 h after application of the adenoviruses as indicated and immunostained for BrdU, troponin I (Trop I), and DAPI. Scale bars: 50 μm; *$P = 0.0169$.

D   Representative pictures and quantification of cardiomyocytes positive for phospho-histone H3 (pH3) in fetal cardiomyocytes 48 h after application of the adenoviruses as indicated and immunostained for pH3, troponin I, and DAPI. Scale bars: 50 μm; *$P = 0.0145$.

E   Representative picture and quantification of cardiomyocyte cell division in fetal cardiomyocytes 48 h after application of the adenoviruses as indicated and immunostained for aurora B (Auro B), troponin I, and DAPI. Scale bar: 50 μm.

F   Representative pictures, single-channel enlargements, and quantification of cardiomyocyte apoptosis in fetal cardiomyocytes 48 h after application of the adenoviruses and after immunostaining for cleaved caspase 3, troponin I, and DAPI. Scale bar: 100 μm.

G   Representative pictures of cardiac explants of the indicated mice embedded in Matrigel shortly after cryoinjury and immunostained for troponin I 7 days later. Quantification of outgrowing cardiomyocytes is shown on the right. The number within bars indicates the number of mice analyzed in that particular group. Scale bars: 50 μm. **$P = 0.0037$.

H   Representative pictures of cardiac explants like in (G), stained for pH3 in addition. The quantification is shown on the right. Scale bars: 50 μm. *$P = 0.0215$.

Data information: (B–H) The number in the bars indicates the number of cell culture plates analyzed. All data are expressed as mean ± SEM. Unpaired Student's *t*-test (C–H) and one-way ANOVA with Sidak's multiple comparisons test (B) were used to compare groups.
Source data are available online for this figure.

between control and CM-G4-KO mice (Fig 5B and C). IL-13 also induced a marked increase in cardiomyocyte cell cycle activity and mitosis in CM-G4-KO mice, but not in control mice after cryoinjury (Fig 5D), and it did not affect capillary angiogenesis or the formation of αSMA-positive small conductance vessels (Fig 5E and F). Since the transcription factor STAT6 is an essential downstream mediator of the proliferative response in cardiomyocytes and since

STAT6 levels were shown to increase upon chronic IL-13 treatment in cardiac myocytes and other cells (Takeda *et al*, 1996; O'Meara *et al*, 2015), we assessed myocardial STAT6 levels in our mice after cryoinfarction and IL-13 or PBS treatment. As shown in Fig 5G, STAT6 protein levels were reduced in the hearts of CM-G4-KO versus control mice with PBS administration, but significantly increased in these mice due to IL-13 treatment. Interestingly, IL-13

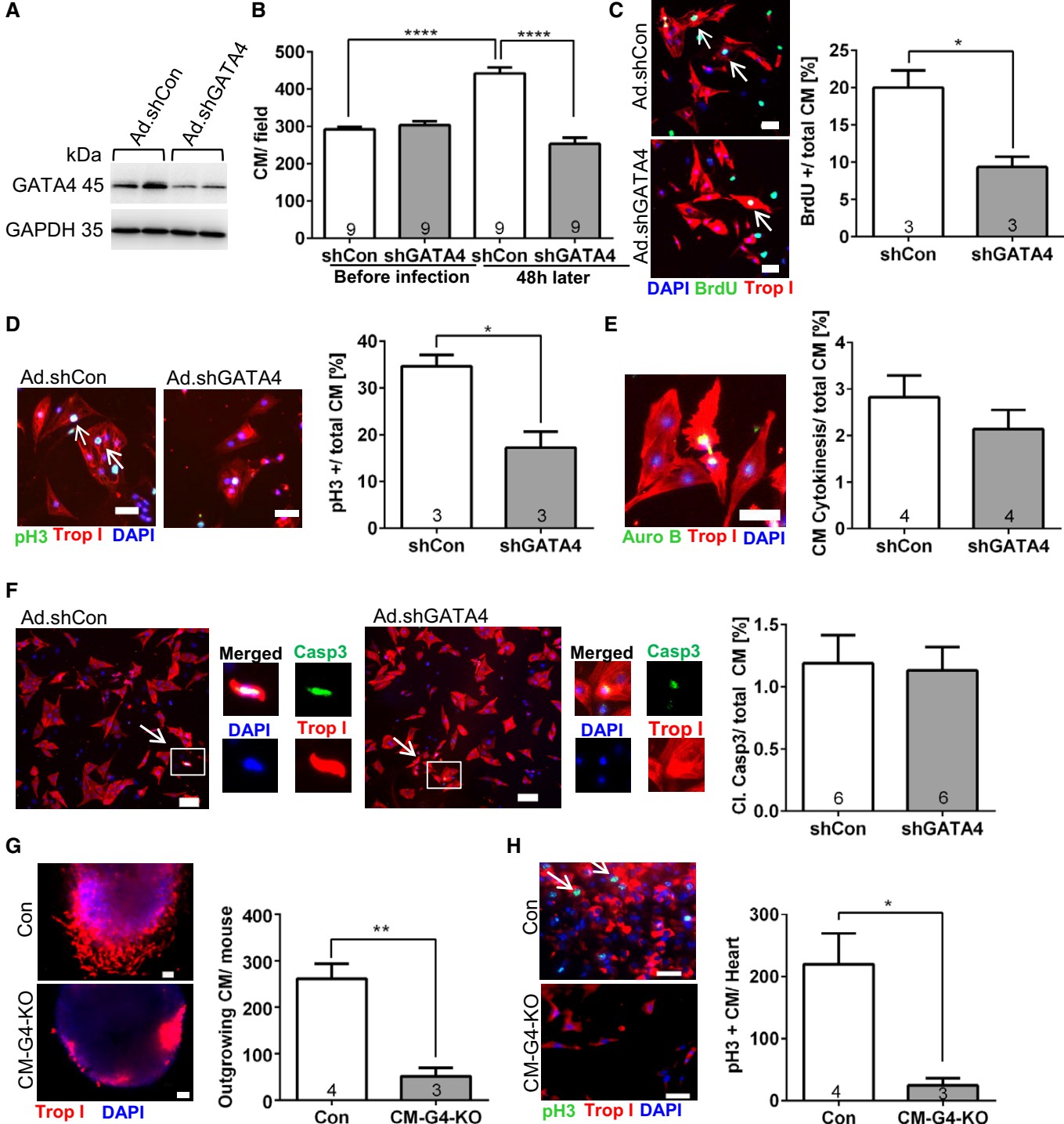

**Figure 3.**

administration also led to an increased myocardial cyclin A2 protein abundance (especially in CM-G4-KO mice) as well as of cenpa (in both control and CM-G4-KO mice) after cryoinjury, suggesting that it indeed might act as an upstream regulator of multiple cardiomyocyte cell cycle-related genes (Fig EV4H and I).

## Discussion

Zebrafish are capable of complete cardiac regeneration after resection of up to 20% of their apical myocardium or cryoinfarction throughout their whole life (Jopling *et al*, 2010). In contrast, mammals (the data are mainly from mice) can only effectively regenerate the heart during embryonic development or shortly after birth until P7, when myocyte proliferation ceases (Porrello *et al*, 2011). Indeed, the expression of cell cycle genes in the heart winds down within the first postnatal week, although transcriptional mechanisms of this phenomenon have remained elusive (Soonpaa *et al*, 1996). We demonstrated in this study that cardiac GATA4 becomes strongly downregulated at P7 and to a lesser extent also after cryoinjury in mice, whereas in zebrafish GATA4 expression is massively upregulated in cardiomyocytes in response to cardiac injury to enable regeneration.

GATA4 downregulation predisposes the neonatal mouse heart to defective regeneration as demonstrated by an increased scar size in the cardiomyocyte-specific *Gata4* knockout mice 7, 21, and 60 days after cryoinjury. Re-expression of GATA4 right after the induction of cryoinjury at P0 reduced the scar size in the CM-G4-KO mice toward control levels, indicating direct dependence of the myocardial scar size on GATA4 abundance and excluding any secondary effects due to preformed developmental abnormalities. Along similar lines, elevation of myocardial GATA4 levels by adenoviral gene transfer after P7 in wild-type mice strongly improved cardiac regeneration, indicating that postnatal GATA4 downregulation is indeed of crucial pathophysiological importance for the impaired regenerative ability of the heart after the first week of life.

We used cryoinjury as model to study neonatal heart regeneration, because it generates highly reproducible myocardial lesions (Jesty *et al*, 2012; Polizzotti *et al*, 2015, 2016), as opposed to cardiac apex resection or LAD ligation, which trigger lesions more

variable in size. In addition, apex resection and LAD ligation—although very useful as models for basic research—might be less relevant with regard to human disease as myocardial amputation does not happen in patients and coronary artery blockage only rarely occurs in children (Polizzotti *et al*, 2016). While cardiac cryoinjury is certainly also not found in patients, the pathological changes it induces, such as inflammation, scar formation, and reduced or unchanged levels of cardiomyocyte mitosis and cell division, were also reported in pediatric patients with heart disease (e.g., tetralogy of Fallot) (Wald *et al*, 2009; Polizzotti *et al*, 2015). Similar to what was previously shown, we did not find complete cardiac regeneration after cryoinjury, since a very small scar was still detectable 60 days later in control mice (Babu-Narayan *et al*, 2006; Jesty *et al*, 2012; Darehzereshki *et al*, 2015; Polizzotti *et al*, 2015). In addition, also as previously reported, we did not detect any change in cardiomyocyte mitosis between control mice with cryoinjury or sham surgery (Darehzereshki *et al*, 2015). In contrast to the other studies, we only found a small non-significant increase in capillary density and we did not detect epicardial activation as response to cryoinjury (Darehzereshki *et al*, 2015). These differences might be attributable to the particular mouse strains used or small variations in the cryoinjury procedure, which in our case produced a consistent non-transmural scar.

How does cardiomyocyte GATA4 promote cardiac regeneration? We excluded enhanced activation of the epicardium, increased cardiomyocyte hypertrophy as well as increased recruitment of macrophages as reasons for the GATA4 effects. Previously, we have identified cardiomyocyte GATA4 as positive regulator of cardiac angiogenesis, and indeed, CM-G4-KO mice showed a reduced capillary density after cryoinjury in the myocardium adjacent to the injury site in this study (Heineke *et al*, 2007). At least in part, this might be the consequence of reduced myocardial *Vegfa* expression in the CM-G4-KO mice, which is in accordance with our previous findings showing *Vegfa* as direct GATA4 target in cardiomyocytes. In zebrafish, inhibition of angiogenesis blunts myocardial regeneration and similar results were obtained in neonatal mice (Lepilina *et al*, 2006; Aurora *et al*, 2014). Hence, reduced angiogenesis in the CM-G4-KO mice likely contributed to the defective regenerative response in these mice. However, since reduced or enhanced GATA4 expression also blunted or increased proliferation in isolated

---

**Figure 4.   GATA4 overexpression triggers cardiomyocyte proliferation *in vitro* and improves myocardial regeneration at P7 *in vivo*.**

A   Representative immunoblot for GATA4 and GAPDH from neonatal cardiomyocytes treated with control (Ad.Con) or GATA4-expressing adenovirus (Ad.GATA4).

B   Quantification of BrdU incorporation, pH3 labeling, and cytokinesis in neonatal cardiomyocytes treated as shown. *P* = 0.0272 for BrdU incorporation, *P* = 0.0329 for pH3 labeling, *P* = 0.0217 for cytokinesis.

C   Immunoblot for GATA4 and GAPDH of mouse hearts treated as shown in the experimental timescale above.

D   A quantification of the immunoblot from (C) is shown; **P* = 0.0062.

E   Representative pictures of myocardial sections immunostained for the indicated proteins from mice 7 days after cryoinjury and treatment with Ad.Con or Ad.GATA4. The quantification of cardiomyocytes (CM) staining positive for GATA4 vs. total cardiomyocytes is shown as bar graph (E'). Scale bars: 20 μm. ****P* < 0.0001.

F   Representative Sirius red-stained heart sections of mice 7 days after cryoinjury (performed at P7) and treated with adenoviruses as indicated. The scar is displayed in red (arrows). Scale bar: 500 μm. The quantification of left ventricular (LV) scar area of mice 7 days after cryoinjury is shown as bar graph (F'). *P* = 0.0135.

G   Representative picture of a myocardial section immunostained for the indicated proteins 7 days after cryoinjury. Scale bar: 10 μm. Quantification of Ki67-positive cardiomyocytes in these mice (G'). **P* = 0.0023.

H   Quantification of pH3-positive cardiomyocytes and cardiomyocyte (CM) cytokinesis in mice 7 days after cryoinjury; *P* = 0.0264, ***P* = 0.0006.

I   Quantification of the capillary/cardiomyocyte ratio in mice 7 days after cryoinjury. ***P* = 0.0002.

Data information: (B, D–I) The number within bars indicates the number of mice analyzed in that particular group. All data are expressed as mean ± SEM. Unpaired Student's *t*-test (B, E–I) and one-way ANOVA with Sidak's multiple comparisons test (D) were used to compare groups.
Source data are available online for this figure.

pure cardiomyocytes in culture (without endothelial cells), an intrinsic effect of GATA4 on cardiomyocyte proliferation also must have contributed to GATA4-dependent regeneration. Indeed, GATA4 is known to regulate the expression of cell cycle genes in the developing myocardium and we found here that the expression of *Ccna2* (encoding cyclin A2), *Ccne1* (encoding cyclin E1), *Cdk4* (encoding

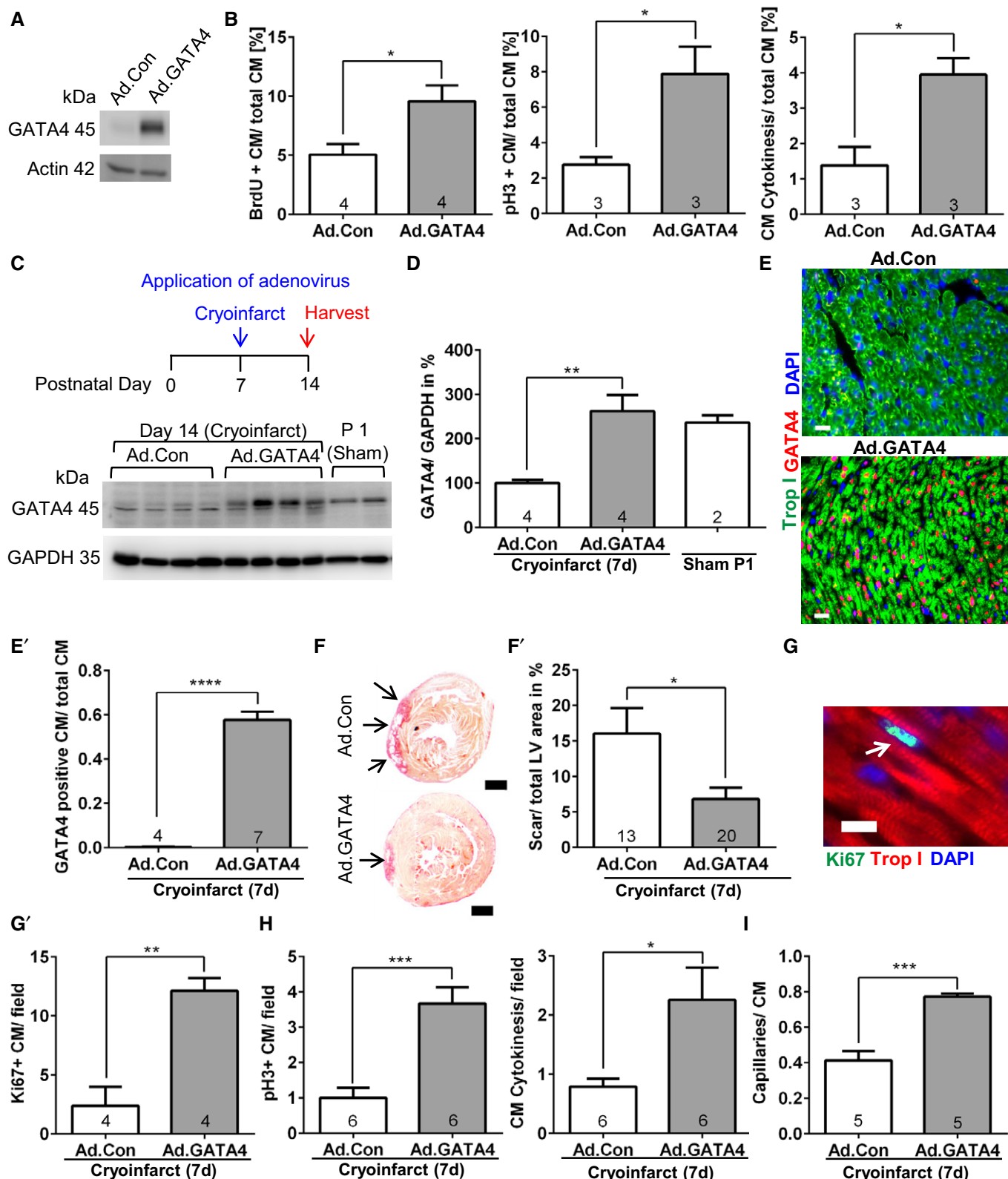

**Figure 4.**

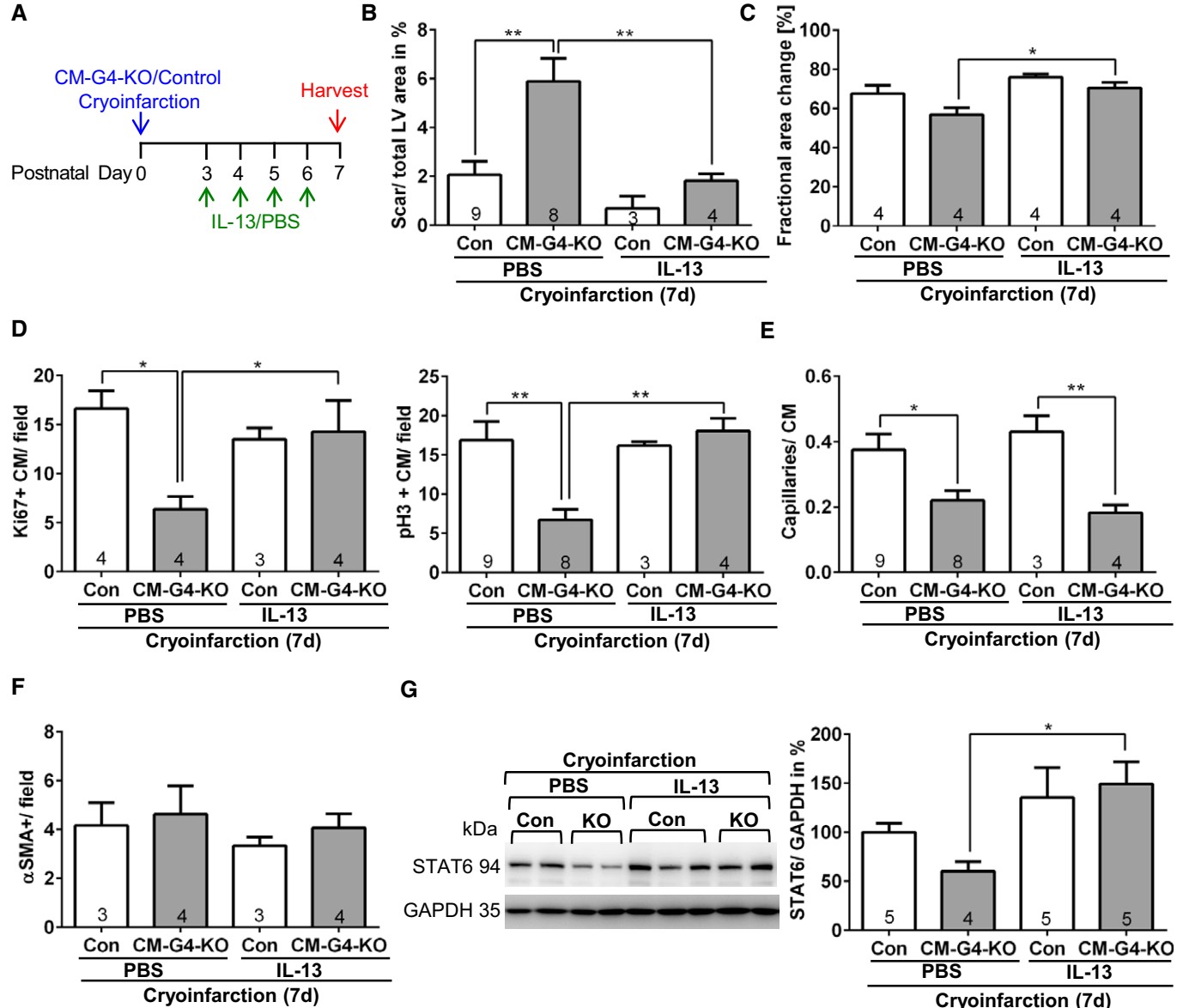

**Figure 5. Systemic treatment with recombinant IL-13 rescued the regenerative defects in CM-G4-KO mice.**

A  Depiction of the experimental timescale.

B  Quantification of left ventricular (LV) scar area of mice 7 days after cryoinjury and treated as indicated. **$P$ = 0.0017 between Con and CM-G4-KO after PBS treatment and **$P$ = 0.0078 between CM-G4-KO mice with PBS or IL-13 treatment.

C  Echocardiographic analysis of left ventricular systolic function in the indicated mice 7 days after cryoinjury; *$P$ = 0.0218.

D  Quantification of Ki67- or pH3-positive cardiomyocytes in mice treated as described in (A); *$P$ = 0.0106 and **$P$ = 0.0052 between Con and CM-G4-KO after PBS treatment and *$P$ = 0.0439 or **$P$ = 0.0023 between CM-G4-KO mice with PBS or IL-13 treatment.

E  Quantification of capillary density as capillary/cardiomyocyte (CM) ratio in mice treated as described in (A). *$P$ = 0.0365, **$P$ = 0.0031.

F  Quantification of αSMA-positive small conductance vessel (~20–50 μm in diameter) in myocardial sections of the mice treated as indicated.

G  Representative immunoblots of STAT6 and GAPDH from neonatal mouse hearts (KO = CM-G4-KO) treated as indicated. The quantification of the blots is shown on the right. *$P$ = 0.0114.

Data information: (B–G) The number within bars indicates the number of mice analyzed in that particular group. All data are expressed as mean ± SEM. One-way ANOVA with Sidak's multiple comparisons test (B–G) were used to compare groups.

Source data are available online for this figure.

cyclin-dependent kinase 4), *Cdk1,* and the transcription factor *E2f1* at least partially depend on the presence of GATA4, which was already previously shown for *Ccna2* and *Cdk4,* whereby *Cdk4* appeared as direct GATA4 target (Rojas *et al,* 2008). Moreover, we

found here that the expression of *Cenpa*—a critical component of cell cycle machinery that is necessary for proper assembly of the mitotic spindle—as well as the expression of the zinc finger transcription factor 191 gene (*Zfp191*), which was deemed highly

necessary for cell proliferation in the early embryo, was downregulated in CM-G4-KO mice (Li *et al*, 2006; McGregor *et al*, 2014). Accordingly, the CM-G4-KO mice exerted less cardiomyocyte mitosis and cell division after cryoinjury, while GATA4 overexpression enhanced these processes. Proliferation of cardiomyocytes was deemed mainly responsible for cardiac regeneration in neonatal mice and zebrafish, and therefore, we propose here that the positive effect of GATA4 on heart regeneration is primarily a consequence of enhanced myocyte division (Jopling *et al*, 2010; Senyo *et al*, 2013).

While this manuscript was in its final stage of preparation, a different group reported that cardiac regeneration in response to transmural cryoinfarction and apical resection surgery was hampered in cardiomyocyte-specific neonatal *Gata4* knockout mice, which were generated using a doxycycline-responsive, troponin T-dependent cardiomyocyte Cre in combination with the same *Gata4* flox mouse line used in this study (Yu *et al*, 2016). Defective regeneration in the *Gata4* mutant mice was also accompanied by reduced cardiomyocyte cell division and reduced angiogenesis, but —in contrast to our study—with markedly increased cardiomyocyte hypertrophy, which is, however, in contradiction to the previously reported prohypertrophic role of GATA4 under physiological and pathological stimulation (Oka *et al*, 2006). One potential explanation for the difference between the two studies with regard to hypertrophy might be the stronger initial myocardial injury (leading to a transmural scar) and the persistent strong reduction of systolic heart function in *Gata4* mutant mice in the study by Yu *et al* (2016), which might secondarily trigger compensatory hypertrophy, while the cardiac injury in our study was smaller and non-transmural and thus did not induce hypertrophy. Mechanistically, Yu *et al* (2016) attribute the changes in their *Gata4* mutant mice mainly to reduced expression of *Fgf16*, because administration of *Fgf16* with an AAV9 vector rescued defective regeneration in their model. In contrast, we did not observe consistent downregulation of *Fgf16* in our CM-G4-KO mice and GATA4 overexpression—an approach not used by the other group—even induced a strong suppression of *Fgf16*. Since GATA4 overexpression markedly promoted cardiomyocyte proliferation and improved myocardial regeneration, *Fgf16* might not be the main downstream mediator of GATA4 during regeneration, although it clearly promotes cardiac regeneration upon overexpression as shown by Yu *et al* (2016).

We attribute GATA4-mediated regeneration at least in part to occur via IL-13, a pleiotropic T helper-2-type cytokine, which was recently reported as important upstream inducer of mitosis in isolated neonatal rat cardiomyocytes acting through its receptor IL13Ra1 on these cells (O'Meara *et al*, 2015). Interestingly, *Il13* RNA and protein were demonstrated to become strongly induced upon mechanical load or angiotensin II in isolated cardiomyocytes, suggesting it might act in an autocrine manner (Nishimura *et al*, 2008). We detected reduced or increased *Il13* mRNA upon cardiomyocyte-specific *Gata4* deletion or overexpression, respectively, and we found that GATA4 binds to the IL-13 promoter in mouse hearts after cryoinjury, suggesting regulation of *Il13* by GATA4. Accordingly, GATA4 binding and transcriptional activation of the *Il13* promoter has been previously shown in T cells (Pai *et al*, 2008). More importantly, systemic administration of recombinant IL-13 effectively reduced the myocardial scar size after cryoinjury in CM-G4-KO mice and this was likely due to IL-13-triggered enhanced cardiomyocyte mitosis, whereas only minor

effects of IL-13 were found in control mice with regard to scar formation. IL-13 might act at least in part via increasing myocardial STAT6 levels, which was shown to be a crucial mediator of IL-13-dependent cardiomyocyte proliferation *in vitro* (O'Meara *et al*, 2015). Since IL-13 had no effect on myocardial capillary density after cryoinjury, but still promoted regeneration, one might infer that an increase in myocyte proliferation is more important than angiogenesis during myocardial restitution and that both processes are not directly interdependent, but clearly further studies are needed in this regard. As another note of caution, we would like to point out that we do not attribute the entirety of GATA4-dependent regenerative defects to IL-13, since other target genes of this transcription factor (such as *Ccna2*, *Cenpa*, *Vegfa*, *Igf2r*) will certainly contribute, although some of these genes (such as *Cenpa*) might be upregulated when IL-13 is administered in a therapeutic dose as we show in this study.

As conclusion, a gene-therapy approach to re-express GATA4 or protein therapy with IL-13 might be evaluated as therapeutic strategies to enhance myocardial regeneration in pediatric patients with heart disease or even in adult patients after MI in the future.

## Materials and Methods

### Experimental animals

The mice with cardiomyocyte-specific *Gata4* deletion (CM-G4-KO: Tg(*β-MHC-Cre*);*Gata4^{flox/flox}*) were previously described and were maintained on a mixed SV129/CD1 background (Oka *et al*, 2006; Heineke *et al*, 2007). Littermate *Gata4^{flox/flox}* mice were used as control mice as described before (Oka *et al*, 2006; Heineke *et al*, 2007). Wild-type mice (WT) with and without the β-MHC-Cre transgene (WT-Cre; Tg(*β-MHC-Cre*)) were kept on the same mixed SV129/CD1 background. ICR-CD1 mice were obtained from Charles River Laboratories. Male and female neonatal mice were equally used throughout the study. The animals had free access to water and a standard diet and were maintained on a 12-h light and dark cycle at a room temperature of 22 ± 2°C. The number and age of mice used is indicated in each figure. All animal procedures described in this study were approved by the local state authorities (the Lower Saxony State Office for Consumer Protection and Food Safety, Germany, file number: 33.12-42502-04-11/0488).

### Cryoinfarction

Neonatal mice were anesthetized by cooling on ice for 2 min, as described (Porrello *et al*, 2011). A left lateral thoracotomy was conducted at the fourth intercostal space by blunt dissection of intercostal muscles after incision of the skin. A cryoprobe with a tip diameter of 0.8 mm was cooled in liquid nitrogen for 2 min and was applied to the heart (left ventricular free wall) of mice at their day of birth (P0) for exactly 3 s. For 7-day-old mice, a cryoprobe with a diameter of 2 mm was used and the probe was applied to the heart for 1 s. Subsequently, the wound was closed by using skin adhesive. The total surgery time per mouse was 1–2 min. For sham surgery, the lateral thoracotomy was conducted, and the skin was closed, without application of the cryoprobe. After surgery, the pups

were warmed up by a heating lamp and put back into the parents' cage.

## Recombinant adenovirus

Recombinant Ad.Control (expressing β-galactosidase) and Ad.GATA4 adenovirus have been described before (Heineke *et al*, 2007). Adenoviruses expressing a control shRNA (Ad.shCon) or a shRNA directed against rat *Gata4* (Ad.shGATA4) were also previously described and were a generous gift from Q. Liang (Kobayashi *et al*, 2007).

## Adenoviral application in mice

For *in vivo* application, Ad.Control and Ad.GATA4 adenoviruses were purified by $CsCl_2$ density gradient and subsequent removal of $CsCl_2$ by dialysis. Cryoinjury was induced in ICR-CD1 mice at P7 as described above and a total dose of 10 μl ($1 \times 10^8$ pfu) of Ad.Control or Ad.GATA4 was directly injected into the myocardium adjacent to the injury site. In mice at P0, 2 μl ($1 \times 10^6$ pfu) was injected into the myocardium of CM-G4-KO and control mice adjacent to the injury site.

## IL-13 application in mice

Recombinant murine IL-13 (Peprotech, 210-13) was administered intraperitoneally, 3 (200 ng), 4 (200 ng), 5 (400 ng), and 6 (400 ng) days after cryoinjury and mice were sacrificed 7 days after injury.

## Transthoracic echocardiography

For echocardiography, mice were anaesthetized with 0.5–1.0% isoflurane and placed on a heating pad to maintain body temperature, as described (Zwadlo *et al*, 2015). Non-invasive, echocardiographic parameters were measured with a linear 30-MHz transducer (Vevo 770, Visualsonics).

## Primary rat cardiomyocytes

Neonatal (1–3 day old) or fetal (isolated from rat hearts at embryonic day E17) cardiomyocytes were isolated as previously described (Zwadlo *et al*, 2015). Neonatal cardiomyocytes were infected with 50 MOI of Ad.Control or Ad.GATA4 on the day after isolation and subsequently cultured with 0.5% fetal bovine serum (FBS)-containing Medium 199 including 1% L-glutamine and 1% penicillin/streptomycin for 48 h. Fetal rat cardiomyocytes were infected with 50 MOI of Ad.shCon or Ad.shGATA4 and cultured subsequently for 48 h in Medium 199 containing 15% FBS. BrdU labeling reagent (Invitrogen) was added into the media at a dilution of 1:100 for 24 h before the cells were fixed.

## Explant tissue culture

Cryoinjury was performed in 1-day-old CM-G4-KO or control mice as described above. Three hours later, the mice were sacrificed; the heart (without atria) was removed and cut into small pieces. One piece of myocardial tissue was embedded in 25 μl of liquid Matrigel (BD Biosciences) within each well of a 48-well plate. After

solidification of the Matrigel, 0.5 ml of Medium 199 containing 15% FCS, 1% L-glutamine and 1% penicillin/streptomycin was added to each well. The culture media were changed every 2–3 days, and after 7 days, the explants were fixed with 4% paraformaldehyde (PFA) in PBS for 10 min and permeabilized with 0.2% Triton X before immunostaining against troponin I was performed. All troponin I-positive cells outgrowing from all explants per heart were quantified. Explants were also separately co-stained for pH3 and troponin I as well as DAPI, and in this experiment, only pH3-positive cardiomyocytes were taken into account.

## Immunofluorescence analysis

Cryosections with 7 μm thickness were prepared and immunostained using standard techniques. The following antibodies were used for immunostaining: anti-pH3 (serine 10, Millipore, 06-570, diluted 1:200), anti-BrdU (coupled with Alexa Fluor 488, Invitrogen, 03-3900, diluted 1:25), anti-Ki67 (Abcam, 15580, diluted 1:100), anti-CD31 (BD Biosciences, 550274, diluted 1:50), anti-αSMA (Sigma, c6198, diluted 1:200), anti-GATA4 (Santa Cruz, sc-1237, diluted 1:50), anti-troponin I (Santa Cruz, sc-15368 or sc-8118, diluted 1:50), anti-F4/80 (Abcam, ab15694, diluted 1:100), troponin T (Abcam, ab8295, diluted 1:200), cleaved caspase 3 (Cell Signaling 9661, diluted 1:50), aurora B (Sigma-Aldrich A5102, diluted 1:100), wt-1 (Santa Cruz sc-192, diluted 1:50), CD3 (Abcam, ab16669, diluted 1:50), and c-kit (Santa Cruz sc-168, diluted 1:50). The secondary antibody (1:200, coupled to Alexa Fluor dyes from Invitrogen) was incubated with WGA-TRITC/FITC (1:50, Sigma-Aldrich), when required. Mounting solution with DAPI was used (Roche). Immunofluorescence analysis of tissue sections was performed using a confocal microscope (Leica DM IRB with a TCS SP2 AOBS scan head).

For assessment of cardiac hypertrophy, three sections per mouse were immunostained for WGA and the cardiomyocyte area was measured in three high-power fields (at 400× magnification) per section in each section and the average of all measured cardiomyocytes per mouse was calculated.

## Histological analysis and assessment of infarct size

For histological analysis, the hearts were cut transversely at the middle between apex and cardiac base, and the two pieces were embedded in paraffin with the ventricular opening facing toward the surface of the paraffin block. Transversal sections of 7 μm thickness were prepared. Around 60 sections were cut from each heart, starting at the middle region between apex and base. After paraffin removal, every 5th slide was stained (each containing at least three heart sections, thus altogether around 12 sections per heart) with the Sirius red staining method using standard procedures. From the 12 stained sections per heart, the section with the largest infarct expansion (stained red for collagen enrichment) was chosen and the infarcted versus the total myocardial area of the left ventricular myocardium was quantified using ImageJ software.

## Staining for BrdU incorporation

Cardiomyocytes were fixed with 100% ethanol and permeabilized with 0.2% Tween-20. The DNA was denatured by incubation with

## The paper explained

### Problem

Adult mammals cannot regenerate contractile myocardium after injury such as ischemia during myocardial infarction or mechanical strain during pathological overload, for example, as consequence of inherited or acquired disorders of the cardiac valves. It has recently been suggested, however, that mice (and possibly humans) retain at least some capability of myocardial regeneration within the first one or two days after birth until postnatal day (P) 7. Loss of cardiac regenerative ability at that time is mainly based on the fact that cardiomyocytes lose their ability to undergo mitosis and cell division. Upstream regulatory mechanisms that trigger the postnatal loss of regenerative myocardial capacity, however, have remained largely unknown. Since the inability to regenerate leads to myocardial scar formation and often ultimately to the development of heart failure—a medical condition with high morbidity and mortality—identification of these key regulatory mechanisms might lead to the development of therapeutic approaches that foster cardiac regeneration and thereby prevent heart failure.

### Results

We found in this study that the cardiomyocyte transcription factor GATA4, which is crucial to promote prenatal cardiac development, becomes strongly downregulated in the mouse myocardium between postnatal days 1 and 7, in parallel with the loss of myocardial regenerative capacity. When we increased the myocardial levels of GATA4 in mice at P7 toward those seen at P1 by adenoviral gene transfer, cardiac regeneration was markedly improved after experimental myocardial cryoinjury, as evident by reduced scar formation, increased cardiomyocyte mitosis, cell division, and angiogenesis (denotes the formation of new blood vessels). In turn, when we analyzed mice with cardiomyocyte-specific *Gata4* knockout (with strongly reduced cardiac GATA4 levels at P1), these mice exerted the opposite—markedly decreased cardiac regeneration after cryoinjury. Decreased regeneration in these mice was accompanied by the reduced expression of selected genes with putative pro-regenerative function. One of the downregulated genes was encoding for the cytokine IL-13. Interestingly, systemic administration of IL-13 strongly promoted cardiac regeneration in cardiomyocyte-specific *Gata4* knockout mice, indicating that its reduced cardiac expression is important for impaired myocardial regeneration in these mice.

### Impact

Reduced myocardial GATA4 levels contribute to impaired cardiac regeneration in mice after postnatal day 7, and future studies need to verify whether this mechanism also plays a role in humans. If this is the case, enhancement of cardiac GATA4 levels or administration of IL-13 might be evaluated as therapeutic strategy to improve myocardial regeneration in heart disease.

0.1 M sodium borate. The anti-BrdU antibody was diluted 1:25 and incubated overnight. A counterstaining for troponin I was performed.

## Immunoblot analysis

Immunoblot analysis was performed from isolated cardiac myocytes or mouse hearts using anti-GATA4 (Santa Cruz, sc-1237, diluted 1:500), STAT6 (Abcam, ab44718, diluted 1:500), cyclin A2 (Abcam ab38, diluted 1:400), cenpa (Cell Signaling C51A7, diluted 1:1,000), and anti-GAPDH (Fitzgerald, 10R-G109a, diluted 1:6,000) or anti-actin (Sigma, A2066, diluted 1:10,000) antibodies following

standard procedures. Densitometry analysis was conducted with Quantity One software (Bio-Rad).

## Quantitative real-time PCR

Total RNA was extracted using TriFast (Peqlab). cDNA was synthesized from 1 µg RNA using Maxima H Minus First Strand cDNA Synthesis Kit (Thermo Fisher Scientific), and quantitative real-time PCR was performed using SYBR Green (Thermo Fisher Scientific) on a MX4000 multiplex QPCR system (Stratagene). Transcript quantities were normalized to *Gapdh* mRNA. Primer sequences are given in the Appendix Table S3.

## *In situ* hybridization for epicardial markers

*In situ* hybridization analysis of mouse hearts 7 days after cryoinjury or sham surgery on paraffin sections with digoxigenin-labeled antisense riboprobes was performed as previously described (Moorman *et al*, 2001).

## Chromatin immunoprecipitation assay

The procedure was conducted from hearts of 7-day-old ICR/CD-1 mice by using the ChIP Assay kit following the manufacturer's instructions (Millipore, 17-295). The samples were incubated for immunoprecipitation with 4 µg GATA4 antibody (Santa Cruz, sc-1237) or 4 µg anti-goat IgG (Santa Cruz, sc-2020) overnight at 4°C, as described (Heineke *et al*, 2007). qPCR was performed using primers specific for the IL-13 promoter in part (for primer 1) as previously described and as listed in the Appendix Table S3 (Pai *et al*, 2008).

## Statistical analysis

Statistical analysis was performed using Prism 6 (GraphPad Software). Data are shown as mean ± standard error of the mean (s.e.m.). Sample size was chosen as a result of previous experience regarding data variability in similar models. No statistical method was used to predetermine sample size. All experiments were carried out in at least three biological replicates. The experiments were not randomized. The investigators were blinded for mouse genotype and treatment during surgeries, echocardiography, organ weight determination, and all histological and immunofluorescence quantifications. Premature death was a pre-established criterion for exclusion from an ongoing mouse experiment. The variance was comparable between groups and normality was assumed. Multiple groups were compared by one-way repeated-measures analysis of variance (ANOVA) followed by the Sidak's multiple comparisons test or by unpaired, two-sided Student's *t*-test when comparing two experimental groups. Differences were considered significant when $P < 0.05$.

**Expanded View** for this article is available online.

## Acknowledgements

This study was supported by the Deutsche Forschungsgemeinschaft through the Cluster of Excellence REBIRTH (EXC 62/3), the Heisenberg Program (HE 3658/6-1 and HE 3658/6-2), and a research grant (HE 3658/11-1).

## Author contributions

MMM and JH designed the study, planned all experiments, and analyzed the data. MMM, BK, AG, NF, MK-K, AG, US, and CR performed experiments. QL contributed critical reagents. AK, KCW, and JB gave advice for the project and critically revised the manuscript. JH wrote the manuscript and supervised the study. All authors read and approved the manuscript.

## Conflict of interest

The authors declare that they have no conflict of interest.

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
