## [Review Process File · EMBO Molecular Medicine]

The transcription factor GATA4 promotes myocardial regeneration in neonatal mice

Mona Malek Mohammadi, Badder Kattih, Andrea Grund, Natali Froese, Mortimer Korf-Klingebiel, Anna Gigina, Ulrike Schrameck, Carsten Rudat, Qiangrong Liang, Andreas Kispert, Kai C. Wollert, Johann Bauersachs, and Joerg Heineke

Corresponding author: Joerg Heineke, Medizinische Hochschule Hannover

Review timeline:

Submission date:	13 May 2016
Editorial Decision:	20 June 2016
Revision received:	07 October 2016
Editorial Decision:	11 November 2016
Revision received:	25 November 2016
Accepted:	30 November 2016

Transaction Report:

Editor: Roberto Buccione

1st Editorial Decision

20 June 2016

Thank you for the submission of your manuscript to EMBO Molecular Medicine. We have now heard back from the Reviewers whom we asked to evaluate your manuscript.

We are sorry that it has taken longer than usual to get back to you on your manuscript. In this case we experienced difficulties in securing appropriate reviewers and then obtaining their evaluations in a timely manner. Further to this, I wished to discuss the evaluations further with my colleagues

As you will see, while Reviewer 1 is more positive, the issue of novelty not surprisingly emerges as a major one (Reviewers 2 and 3). Clearly it appears that you have not succeeded in convincing the expert reviewers of your claims, due to the lack of sufficient experimental support.

I do note however, that Reviewers 2 and 3 do recognize the interest and also potential novelty of your work, pending direct experimental support. In conclusion, while publication of the paper cannot be considered at this stage, and after further discussion with my colleagues and reviewer cross-commenting, we have decided to give you the opportunity revise your manuscript.

We are thus prepared to consider a substantially revised submission, with the understanding that the Reviewers' concerns must be addressed with additional experimental data where appropriate and that acceptance of the manuscript will entail a second round of review. I suggest that you address each point in detail. The overall aim is to significantly upgrade the relevance and usefulness of the dataset and its news value, which of course are all of paramount importance for our title.

I understand that if you do not have the required data available at least in part, to address the above, this might entail a significant amount of time, additional work and experimentation and might be technically challenging, I would therefore understand if you chose to rather seek publication elsewhere at this stage. Should you do so, we would welcome a message to this effect.

It is important that you consider that it is EMBO Molecular Medicine policy to allow a single round of revision only and that, therefore, acceptance or rejection of the manuscript will depend on the completeness of your responses included in the next, final version of the manuscript.

As you know, EMBO Molecular Medicine has a "scooping protection" policy, whereby similar findings that are published by others during review or revision are not a criterion for rejection. However, I do ask you to get in touch with us after three months if you have not completed your revision, to update us on the status. Please also contact us as soon as possible if similar work is published elsewhere.

Finally, please note that EMBO Molecular Medicine now requires a complete author checklist (<http://embomolmed.embopress.org/authorguide#editorial3>) to be submitted with all revised manuscripts. Provision of the author checklist is mandatory at revision stage; The checklist is designed to enhance and standardize reporting of key information in research papers and to support reanalysis and repetition of experiments by the community. The list covers key information for figure panels and captions and focuses on statistics, the reporting of reagents, animal models and human subject-derived data, as well as guidance to optimise data accessibility.

Please note that we now mandate that all corresponding authors list an ORCID digital identifier. You may do so through our web platform upon submission and the procedure takes <90 seconds to complete. We also encourage co-authors to supply an ORCID identifier, which will be linked to their name for unambiguous name identification.

***** Reviewer's comments *****

Referee #1

Comments on Novelty/Model System

Mohammadi and colleagues provide an elegant study and identify GATA4 as an critical driver of myocardial regeneration in neonatal mice. Since the discovery that neonatal hearts can regenerate it is now possible to search for critical factors. The authors set out to study the role of GATA4 and use both gain and loss of function. Moreover, they identify IL13 as one of the critical downstream cytokines. The manuscript is well written and the experimentation is technically sound, the data clear and convincing. I have one a minor comment:

1. The authors correctly state that IL13 "at least in part" to contribute to GATA4 pro-regenerative effects. But can they discuss in more detail the potential contribution of the other factors and - as a limitation - that administration of IL13 may also act independent of GATA4. The more convincing experiment (that I admit is beyond of the current study) would be to prevent adenovirus GATA4-induced regeneration in IL13-deficient mice.

Referee #2

Mohammadi and colleagues examined the impact of cardiomyocyte GATA4 for neonatal heart regeneration and observed that cardiac GATA4 protein is abundant in mice shortly after birth, but reduces dramatically at P7, when also the regenerative capacity of the heart is diminished. Cardiomyocyte-specific deletion of GATA4 at P0 results in impaired cardiac regeneration after myocardial cryoinjury, whereas replenishment of GATA4 at P7 led to improved cardiac regeneration. They suggest IL13 is a strong inducer of this response. Although the study is well conducted and timely, thereby using several experimental approaches to provide evidence towards

the role of GATA4 in CM proliferation, novelty is hampered by previous work which is not an important issue if the authors could give a more detailed mechanistic explanation of their observations since they could not confirm previous work.

Major

- % of GATA4 positive CM is relative low (<0.1%, figure 1E), which is in line with previous hints but for Fig 4C and E, please provide % of positive cells, as indicated and detected in figure 1E
- Small decrease in ejection fraction is observed in relative small number of animals, but how relevant is this. Please provide more functional regional measurements in the myocardium to demonstrate a consequence of GATA4 loss. Is there a difference in CM mass loss, acutely after cryo-injury. Since the scar mass is dropping in time in both groups, a clear initial difference could explain the timely response.
- Are GATA4 areas and markers of fig4C and E in the adjacent injury area, these pictures should be provided, since this fits the rationale of the authors.
- Figure 3G is an assay that is usually mainly driven by cellular migration and less by cell proliferation. This hints as well towards defects in the GATA4 animals for migration/angiogenesis etc, which could further explain the observations in these animals and the reduction in scar mass contraction as indicated in figure 1.
- What is the effect on inflammatory influx in these animals and due to their treatments, as indicated in their manuscript macrophages are key effectors in this response but also they suggest a role of Tcells. Although Suppl figure shows no effect on F4/80 stainings, further details on subtypes are needed to exclude this option.
- How does IL13 work, since the experimental approach suggest systemic activation of cell types? Also the observations in the GATA4 KD experiments suggest that intercellular signaling is somewhat involved and that this cross-talk might be involved.

Minor

- Figure 4E seems to have larger scar areas as in Figure 1, please explain? Figure 5B has significant lower areas again?
- Figure 4F need as well %of cells.
- It appeared that number of capillaries in GATA4 KD animals is somewhat lower at the initial stage, but currently underpowered to demonstrate this.
- Figure 3A, why only the number of cells per field? In vitro allows the quantification of all cells?

Referee #3

Comments on Novelty/Model System

See comments below.

This study by Mohammadi and colleagues describes how GATA4 is required for effective cardiac repair and regeneration in neonatal mice, via effects on cardiomyocyte proliferation and angiogenesis. They suggest that the role of GATA4 in neonatal cardiomyocyte proliferation may be mediated by IL-13 as a previously described inducer of mitosis. The authors investigate cryo-injury of neonatal (P0) myocardial-specific GATA4 knock out (CM-G4-KO) mice, adenoviral delivery of shRNAi-mediated GATA4 knockdown and Ad-GATA4 over-expression in neonatal rat cardiomyocytes, over-expression of GATA4 in wild type mice followed by cryo-injury at P14, to reveal enhanced CM proliferation, and recombinant IL-13 treatment to effect rescue of scar size and cardiomyocyte proliferation in the CM-G4-KO mutants.

This is an interesting study in implicating over-expression of GATA4 and/or IL-13 as a potential route for protein or gene therapy to extend the neonatal regenerative window. That said, there is an issue with the novelty of the major finding, of a role for GATA4 in neonatal mouse heart regeneration, given the previously published findings by Yu and colleagues in *Development* earlier this year (Yu et al., 2016 *Development* 143, 936-949). The latter investigated a role for inducible/neonatal loss of myocardial GATA4 in the context of cryo-injury and revealed similar findings of reduced cardiomyocyte proliferation and impaired angiogenesis but with some notable differences (addressed below). In addition, there are issues with the data in their current form that need addressing and further analyses required to fully support the conclusions as they stand:

1. What was the outcome/long term phenotype in the cryo-injured CM-G4-KO mutants?
2. In Figure 1F how was the infarct size assessed? If this was done by histology then significant details on methodology (number of sections analysed, assessment of infarct area etc) are missing and arguably this is not an accurate approach which requires functional imaging.
3. There is no immunofluorescence (IMF) data (with endothelial/smooth muscle markers) to support the reduced coronary angiogenesis/neovascularisation shown quantitatively in Figure 2A; reduced CD31 gene expression is insufficient here.
4. The aurora B-kinase image in Figure 2D is very poor in resolution and it is unclear to where the signal is localised as to be consistent with cytokinesis(?). The accompanying quantitative data reveals a difference of 1 versus 3 +ve cells per field between control and CM-G4-KO hearts, which is prone to significant error and bias. Was the assessor blinded to the genotype during the analysis?
5. Gene expression data (Figure EV3) does not accurately represent changes in cell numbers as suggested for c-kit+ CSCs, cd14-macrophages and wt1-epicardial cells (page 5). This needs to be assessed via IMF/cell counts (EPDCs) and/or FACS (CSCs and macrophages).
6. Figure EV2F reveals no differences in cardiac hypertrophy: how was this assessed? There should be section data presented with staining for CM membranes (eg WGA) and a study of multiple sections, to compensate for variable plane of section which can artificially shorten CMs in any single orientation.
7. The in situ data in Fig. E3VB suggesting no epicardial activation is not convincing. The panels shown represent low resolution images of whole heart sections with relatively high background. This should be accompanied by high (cellular) resolution images focusing on the epicardium across different regions of the heart; qPCR for Wt1, Raldh2, Tcf21 (not Tbx18 given it may be expressed in neonatal myocardium with injury) and/or IMF data. Given the potential finding of a lack of epicardial activation contrasts with previous published findings (Darehzereshki et al., 2015) this needs significant further analyses.
8. The relevance of adenoviral GATA4 knock down and o/e experiments is unclear given the substrate of rat fetal cardiomyocytes(?). These should be carried out on neonatal mouse cardiomyocytes for comparative purposes or preferably in vivo as per the subsequent Ad-GATA4 myocardial injection experiments. In addition, BrdU pulse-chase was employed as part of the fetal rat CM study and it is unclear why this (or EdU) was not utilised for the in vivo assessment of effects on CM cell cycle (S-phase) activity as a more reliable measure than PH3-staining(?).
9. Figure 3F, the quantitative data on levels of apoptosis requires accompanying IMF staining for cleaved caspase 3 to be presented.
10. The rationale for focusing on IL-13 is missing from the results section on page 7. Where is the evidence of elevated IL-13 protein in the hearts of those mice that received ip injections? This needs to be included.
11. The addition of recombinant IL-13 to cryo-injured CM-G4-KO mutants appears to completely rescue both scar size and number of PH3+ CMs (Figure 5B, C). How is this feasible given the multiple downstream targets of GATA4 and documented mis-regulation of other factors such as Vegfa and Igf2r? It is hard to reconcile complete "rescue" of these parameters via a single factor(?) and as such this requires further explanation. Is IL-13 a direct target of GATA4 in the heart? Previous interactions were demonstrated in Th1 cells by Pai et al., (2008).
12. The significant differences between this study and that of Yu and co-workers (2016), regarding CM hypertrophy and the regulation of Fgf16 are not adequately explained in the discussion on page 9; especially given that the approach of cardiomyocyte-specific ko and cryo-injury were common to both studies. Why do the authors think they failed to observe a hypertrophic response or consistent down-regulation of Fgf16 in a loss of GATA4 background? Might this be explained by their constitutive CM-KO potentially having subtle effects during development, versus the inducible CM-KO targeted to neonatal stages adopted by Yu et al? How do the authors explain Yu and colleagues observations of partial rescue of regeneration by administration of AAV9-Fgf16 in light of their findings?

1st Revision - authors' response

07 October 2016

The Reviewers have brought up some very good points and we appreciate the opportunity to clarify our research objectives and results. As indicated below, we have checked all the general and specific comments provided by the Reviewers (shown in bold) and have responded to all of them as shown below (shown in normal letters):

Reviewer #1

1. The authors correctly state that IL13 "at least in part" to contribute to GATA4 pro-regenerative effects. But can they discuss in more detail the potential contribution of the other factors and - as a limitation - that administration of IL13 may also act independent of GATA4. The more convincing experiment (that I admit is beyond of the current study) would be to prevent adenovirus GATA4-induced regeneration in IL13-deficient mice.

We greatly appreciate the reviewer's efforts to carefully review the paper and the valuable suggestions offered. Although IL-13 was identified by Richard Lee's group as an upstream regulator of regeneration in neonatal mice (and especially cardiomyocyte proliferation) after heart injury (Meara et al, 2014), we admit other factors (such as cyclin A2, Cenpa or the IGF2 receptor) surely contribute to the GATA4 dependent effects on regeneration. That said, it is still noteworthy that IL-13 seems to be indeed rather upstream as inducer of cardiomyocyte proliferation, since we now show in the revised manuscript that it leads to myocardial upregulation of cyclinA2 and cenpa (new Figure EV4G, H), two cell cycle promoting genes that we also find to be dependent on GATA4.

As requested by this reviewer, we added a note of caution with regard to IL-13 as mediator of GATA4 effects (Discussion, page 10, lines 28-32): "*As another note of caution, we would like to point out that we do not attribute the entirety of GATA4 dependent regenerative defects to IL-13, since other target genes of this transcription factor (such as Ccna2, Cenpa, Vegfa, Igf2r) will certainly contribute, although some of these genes (such as Cenpa) might be upregulated when IL-13 is administered in a therapeutic dose as we show in this study.*"

The suggested experiment is really interesting and we will definitively conduct it in the future. As the reviewer mentioned we also feel that it is unfortunately beyond the scope of this study.

Reviewer #2**Major**

1- % of GATA4 positive CM is relative low (<0.1%, figure 1E), which is in line with previous hints but for Fig 4C and E, please provide % of positive cells, as indicated and detected in figure 1E.

We appreciate the suggestion. To address this comment we stained the heart sections for GATA4 and troponin I. GATA4 positive cardiomyocytes were quantified in Ad.GATA4 and Ad.bgal infected hearts after cryoinfarction. The quantification as well as an example images are added in the new Figure 4D.

2- Small decrease in ejection fraction is observed in relative small number of animals, but how relevant is this. Please provide more functional regional measurements in the myocardium to demonstrate a consequence of GATA4 loss. Is there a difference in CM mass loss, acutely after cryo-injury. Since the scar mass is dropping in time in both groups, a clear initial difference could explain the timely response.

To investigate the amount of CM mass loss directly after cryoinjury, we analyzed neonatal mouse hearts 3 hours after cryoinjury. After embedding the hearts in O.C.T they were cut through the whole organ and every second slide was stained by the TUNEL method to indicate the region of initial cell death (this approach has also been taken by Polizzotti, BD et al, Sci Transl. Med, 2015). As shown in the new Figure 2A in the revised manuscript, the initial injury was similar between control and CM-G4-KO mice. Furthermore, heart weight/ body weight was measured and as shown in the new Figure EV2E was also not different between two groups (also indicating a similar "starting point" in both groups of mice). Thus, since the initial injury was similar between the groups, the differences in cardiac scar size that we observe 3-60 days after injury are most likely attributable to differences in regeneration capacity.

Indeed, we agree with this reviewer that the changes in ejection fraction after injury are only moderate, which however fits to our model of non-transmural myocardial cryoinjury. This is

clinically relevant, since in clinical practice transmural infarctions become less frequent compared to non-transmural infarctions in patients due to early reperfusion therapy.

Since ejection fraction is calculated in our echo machine from the distance of two different points in the left ventricle (LV) in systole vs. diastole, we exchanged this by fractional area change (new Figure 1I), which is calculated from the whole horizontal area of the LV in systole and diastole (and therefore gives a more accurate measure). The fractional area change is determined as follows in detail: LV end-diastolic area (LVEDA) and end-systolic area (LVESA) were recorded. Fractional area change was calculated as $[(LVEDA-LVESA)/LVEDA] \times 100$.

3- Are GATA4 areas and markers of fig4C and E in the adjacent injury area, these pictures should be provided, since this fits the rationale of the authors.

We apologize for not clearly mentioning that in Figure 4C we used total protein, which was isolated from the whole heart. To provide more insight to our experiment, heart sections were immunofluorescence stained for GATA4 and troponin I, pictures were taken from all over the heart and global over-expression of GATA4 was observed after administration of Ad.GATA4 as now added to the new Figure 4D (including quantification).

Global cardiac expression after local administration of an adenovirus in neonatal hearts was previously discovered in a publication by H. Ebel and T. Braun in 2003 (BBRC), and is due to lack of fully developed adaptive immunity in neonatal mice.

4- Figure 3G is an assay that is usually mainly driven by cellular migration and less by cell proliferation. This hints as well towards defects in the GATA4 animals for migration/angiogenesis etc, which could further explain the observations in these animals and the reduction in scar mass contraction as indicated in figure 1.

This is a very good point. To address this and to distinguish between the cardiomyocytes growing out due to proliferation versus just migrating out of the explant we repeated the experiment. This time, the explants and the outgrowing cells were stained for Troponin I, pH3 and DAPI (new Figure 3H). Cardiomyocytes, which were positive for proliferation marker (pH3) were quantified. We found more pH3 (i.e. mitotic) cardiomyocytes in heart explants from WT versus CM-G4-KO hearts. Therefore, cardiomyocytes were likely growing more out of the control explants due to migration (the reviewer is right at this point), but also proliferation of cardiomyocytes. We now acknowledge the likely contribution by increased migration in the results section (page 6, lines 35-38): "*Because also cell migration plays a role in this assay, it is likely that reduced cardiomyocyte migration and reduced mitosis in combination led to diminished cardiomyocyte outgrowth from CM-G4-KO tissue*".

5- What is the effect on inflammatory influx in these animals and due to their treatments, as indicated in their manuscript macrophages are key effectors in this response but also they suggest a role of T cells. Although Suppl figure shows no effect on F4/80 stainings, further details on subtypes are needed to exclude this option.

We now investigated the expression of CD86 as M1 macrophages marker and CD206 as marker for the M2 family of macrophages by qPCR. As it was previously described (Aurora et. al, 2014) M2 macrophages, which promote wound healing, became more abundant in our neonatal mice after myocardial injury, but this occurred to a similar extent in control and CM-G4-KO mice (new Figure EV3A). The expression of CD4 gene as a marker of T-helper cells was also measured, but there was no difference 7 days after cryoinjury between wild-type and knock out mice (revised Figure EV3A).

To study the role of T-cells and their possible role in our model with another method, we performed immunostaining for CD3. Although we could clearly observe T-cells in our positive control, no obvious T-cell staining was detected 7 days after cryoinjury in the heart sections from our mice (revised Figure EV3E).

6- How does IL13 work, since the experimental approach suggest systemic activation of cell types? Also the observations in the GATA4 KD experiments suggest that intercellular signaling is somewhat involved and that this cross-talk might be involved.

This is a very good point. We studied this by repeating the experiment of injecting IL13 and PBS in mice after cryoinjury. Seven days after cryoinjury protein was extracted from the whole heart. We clearly observed that STAT6 protein level, which is a target of IL-13 (Takeda et. al, 1996; O'Meara et al., Circ Res, 2015) is less abundant in our injured GATA4 knock-out mice, but became upregulated in mice injected with IL-13 compared to those injected with PBS. This data is now included in the revised manuscript (new Figure 5E, F). STAT6 was previously identified as essential for cardiomyocyte cell cycle stimulation (O'Meara et al., Circ Res, 2015).

Further analysis showed that cyclin A2 and cenpa protein levels are up-regulated in the mice after IL-13 injection (new Figures EV4G, H), suggesting indeed that IL-13 promotes cell cycle promoting gene expression in the heart.

Minor

7- Figure 4E seems to have larger scar areas as in Figure 1, please explain? Figure 5B has significant lower areas again?

We apologize for not clearly explaining that in Fig4E we are showing the scar 7 days after injury induced in 7-day-old mice, but in Figure 1F we are showing the scar 7 days after cryoinjury performed in newborn mice. In addition the cryoprobe, which was used for 7-day-old mice, was bigger (0.8mm in neonatal and 2 mm in 7 day old mice) than the one used for newborn mice. In Figure 5B the mice were handled every day to inject PBS or IL-13 subcutaneously from day 3-6 after surgery. This might have at least partially contributed to the overall reduced infarct size in Figure 5B vs. 1F. In addition in Figure 5, the experiment was conducted in a more advanced generation of mice compared to the mice in Figure 1, which were operated in the very beginning of the study. Importantly, litter mates were directly compared throughout the study, so the relative difference in myocardial regeneration remained similar through the mouse generations.

8- Figure 4F need as well % of cells.

In line with all the other papers on cardiac regeneration for in vivo (mouse) data, we have also quantified and mentioned pH3 positive cardiomyocytes per field throughout our manuscript (O'Meara et al. 2014, Engel et. al 2005, Porrello, et al.2011, Yu et al 2016 and ...). In vitro data from cell culture, we consistently reported as % of cells. Just as an example, we quantified pH3 positive cardiomyocytes as % of cells from the data shown in 4F(see below), in principal showing that both ways to report pH3 positive cells in the end look very similar:

9- It appeared that number of capillaries in GATA4 KD animals is somewhat lower at the initial stage, but currently underpowered to demonstrate this.

We apologize for making that impression, but that was due to existence of only one extreme value with less capillaries in our sham knock-out mice. We now analyzed and added more mice to both sham groups and with that result we are certain that there is no difference in cardiac capillary density between both genotypes after sham surgery. We have shown the new data as well as the representative image in the new Figure 2B.

10- Figure 3A, why only the number of cells per field? In vitro allows the quantification of all cells?

We now quantified the number of cardiomyocytes before and after infection with Ad.shGATA and Ad.shcontrol before and 48 hours after adenoviral infection. This quantification was done in a defined (i.e. in the same) low magnification microscopic field over time. While the control cardiomyocytes increased in number, this was not the case in the shGATA4 treated cells. This data is now shown in the new Figure 3B. To count the whole plate is really a lot for a 10cm plate.

Reviewer #3

1. What was the outcome/long term phenotype in the cryo-injured CM-G4-KO mutants?

We have now performed echocardiography 2 months after surgery as shown in the new Figure EV1G. The injury we induced throughout our study led to non-transmural scars and as shown (Figure 1H) in 60-day-old mice the scar is less than 1% of the LV, which is not big enough to make a difference in the heart function at that time point in any of the groups studies. To study the long term phenotype, a bigger injury might be needed on neonatal day 1. We have tried to induce a bigger injury at that time point before, however, this triggers a very high mortality in the GATA4 knock-out mice (>50% of the mice died); consequently, we reduced the injury size in our study.

2. In Figure 1F how was the infarct size assessed? If this was done by histology then significant details on methodology (number of sections analyzed, assessment of infarct area etc) are missing and arguably this is not an accurate approach which requires functional imaging.

We describe the method of infarct size determination in the Methods section (page 13, line1-10): *“For histological analysis, the hearts were cut transversely at the middle between apex and cardiac base, and the two pieces were embedded in paraffin with the ventricular opening facing towards the surface of the paraffin block. Transversal sections of 7 μ M thickness were prepared. Around 60 sections were cut from each heart, starting at the middle region between apex and base. After paraffin removal, every 5th slide was stained (each containing at least 3 heart sections, thus altogether around 12 sections per heart) with the Sirius-Red staining method using standard procedures. From the 12 stained sections per heart, the section with the largest infarct expansion (stained red for collagen enrichment) was chosen and the infarcted versus the total myocardial area of the left ventricular myocardium was quantified using ImageJ software.”*

As functional imaging, we employ echocardiography, which shows reduced heart function in control mice after cryoinjury vs. sham, and even more reduced heart function in CM-G4-KO mice after cryoinjury (figure 1I). This is in line with our histological determined scar sizes.

3. There is no immunofluorescence (IMF) data (with endothelial/smooth muscle markers) to support the reduced coronary angiogenesis/neovascularisation shown quantitatively in Figure 2A; reduced CD31 gene expression is insufficient here.

Figure2A (revised Figure 2B) is a quantification of images taken from IMF staining of endothelial cells using CD31 marker and also WGA to visualize the cell membrane of cardiomyocytes. In addition, we have now included representative IMF figures in revised Figure 2B. In Figure EV2, we only showed the expression of the CD31 gene, which is in line with our IMF results.

4. The aurora B-kinase image in Figure 2D is very poor in resolution and it is unclear to where the signal is localised as to be consistent with cytokinesis(?). The accompanying quantitative data reveals a difference of 1 versus 3 +ve cells per field between control and CM-G4-KO hearts, which is prone to significant error and bias. Was the assessor blinded to the genotype during the analysis?

The images were taken with a confocal microscope and show a dividing cardiomyocyte. We have magnified this dividing cell, showing localization of aurora B in the midbody of cytokinesis (new Figure 2D). We hope that it is now acceptable. All our samples were labeled by numbers only (not including genotype or treatment) and the person who acquired and analyzed the images was completely blinded.

5. Gene expression data (Figure EV3) does not accurately represent changes in cell numbers as suggested for c-kit⁺ CSCs, cd14-macrophages and wt1-epicardial cells (page 5). This needs to be assessed via IMF/cell counts (EPDCs) and/or FACS (CSCs and macrophages).

This is a good point. We performed IMF staining for c-kit, wt-1 (epicardial), CD3 (T-cells) and F4/80 for macrophages (new Figure EV3C-E & EV2D) and no differences between control and GATA4 knock-out mice after cryoinjury were observed. qPCR as estimate for the abundance of certain cell types has been used in many studies before (for example by Andersen DC, Stem Cell Reports, 2014), but we agree with this reviewer that these data have to be complemented by histological approaches – as we have done throughout our manuscript.

6. Figure EV2F reveals no differences in cardiac hypertrophy: how was this assessed? There should be section data presented with staining for CM membranes (eg WGA) and a study of multiple sections, to compensate for variable plane of section which can artificially shorten CMs in any single orientation.

We apologize for not explaining this properly in our first version of the manuscript. We now added this important information into the revised manuscript (page 12, lines 38-41): *“For assessment of cardiac hypertrophy, 3 sections per mouse were immunostained for WGA and the cardiomyocyte area was measured in 3 high-power fields (at 400x magnification) per section in each section and the average of all measured cardiomyocytes per mouse was calculated.”* We also now added representative pictures in the revised manuscript (new Figure EV2 G).

7. The in situ data in Fig. E3VB suggesting no epicardial activation is not convincing. The panels shown represent low resolution images of whole heart sections with relatively high background. This should be accompanied by high (cellular) resolution images focusing on the epicardium across different regions of the heart; qPCR for Wt1, Raldh2, Tcf21 (not Tbx18 given it may be expressed in neonatal myocardium with injury) and/or IMF data. Given the potential finding of a lack of epicardial activation contrasts with previous published findings (Darehzereshki et al., 2015) this needs significant further analyses.

We took images with higher magnification as now represented in the new Figure EV3B. As suggested we have performed qPCR for Wt-1, Raldh2 and Tcf21 (revised Figure EV3A) as well as immunofluorescence staining for Wt-1, which is shown in revised Figure EV3-D. All these different approaches failed to reveal epicardial activation in our model.

8. The relevance of adenoviral GATA4 knock down and o/e experiments is unclear given the substrate of rat fetal cardiomyocytes(?). These should be carried out on neonatal mouse cardiomyocytes for comparative purposes or preferably in vivo as per the subsequent Ad-GATA4 myocardial injection experiments. In addition, BrdU pulse-chase was employed as part of the fetal rat CM study and it is unclear why this (or EdU) was not utilised for the in vivo assessment of effects on CM cell cycle (S-phase) activity as a more reliable measure than PH3-staining(?).

One has to keep in mind that BrdU has a very short half-life (almost 2 h) and that proliferation is highly variable in the time after birth (Naqvi et.al, 20014) and that is also different between night and day. This makes for example a once daily injection not a proper way for BrdU administration. Ideal would be administration by osmotic minipumps for continuous release, but 1-day-old mice are too small for that approach. Furthermore, pH3 as mitosis marker might be more relevant, since DNA synthesis (as measured by BrdU incorporation) can happen in cardiomyocytes without subsequent mitosis or cell division. In addition we used aurora B to mark cell division as complementary approach.

Rat cardiomyocytes as well as pH3 and aurora B as proliferation marker have been widely used in the field of cardiac regeneration (O’Meara et al. 2014, Engel et. al 2005, Porrello, et al. 2011..) For further insight into mouse cardiomyocyte proliferation in the absence of GATA4 we used the explant tissue culture system where we cultured CM-G4-KO and control mouse hearts and then stained for the proliferation marker pH3 (new Figure 3H). We found significantly less mitotic cardiomyocytes arising from CM-G4-KO vs. control explants.

9. Figure 3F, the quantitative data on levels of apoptosis requires accompanying IMF staining for cleaved caspase 3 to be presented.

We now provided these images in the new Figure 3F.

10. The rationale for focusing on IL-13 is missing from the results section on page 7. Where is the evidence of elevated IL-13 protein in the hearts of those mice that received ip injections? This needs to be included.

We measured the expression of some highly important regenerative genes, especially those known to influence cardiomyocyte proliferation. IL-13 was recently discovered to play an important role as upstream signaling regulator in regeneration due to the induction of cardiomyocyte proliferation and therefore was among the genes we analyzed (O'Meara et al. 2014). We now better address our rationale for following up on IL-13 (page 7, lines 24-33): *"While the regenerative defects in CM-G4-KO mice after cryoinjury are likely the result of the combined changes in gene-expression triggered directly or indirectly through the loss of Gata4 in cardiomyocytes, we still aimed to elucidate whether compensation of reduced cardiac Il13 mRNA expression in CM-G4-KO mice by exogenous, systemic administration of IL-13 could improve heart regeneration, since protein therapy with recombinant IL-13 might be feasible as therapeutic approach in the future. IL-13 was also recently identified as nodal upstream inducer of cardiomyocyte proliferation through a comprehensive gene-expression screen (O'Meara et al, 2015), and in addition, we found by ChIP assay that GATA4 directly interacts with the Il13 promoter in the myocardium of neonatal wild-type mice after cryoinjury, therefore suggesting direct regulation of Il13 expression by GATA4 (Figure EV4 F)."*

Since injected proteins are only transiently detectable, we studied STAT6 protein levels, which were shown to increase upon chronic IL-13 treatment (Takeda et. al, 1996; O'Meara et. al, 2014). Seven days after cryoinjury, protein was extracted from the whole heart. We clearly observed that STAT6 protein levels are less abundant in our injured GATA4 knock-out mice, but became upregulated in mice injected with IL-13 compared to those injected with PBS. This data is now included in the revised manuscript (new Figure 5E, F). STAT6 was previously identified as essential for cardiomyocyte cell cycle stimulation (O'Meara et al., Circ Res, 2015). Further analysis showed that cyclin A2 and cenpa protein levels are up-regulated in the mice after IL-13 injection (new Figures EV4G, H), suggesting indeed that IL-13 promotes cell cycle promoting gene expression in the heart.

11. The addition of recombinant IL-13 to cryo-injured CM-G4-KO mutants appears to completely rescue both scar size and number of PH3+ CMs (Figure 5B, C). How is this feasible given the multiple downstream targets of GATA4 and documented mis-regulation of other factors such as Vegfa and Igf2r? It is hard to reconcile complete "rescue" of these parameters via a single factor (?) and as such this requires further explanation. Is IL-13 a direct target of GATA4 in the heart? Previous interactions were demonstrated in Th1 cells by Pai et al., (2008).

Administration of IL-13 after injury did indeed not rescue angiogenesis, but induced cardiomyocyte proliferation, which is crucial for regeneration and might be the main reason of lack of regeneration in GATA4 knockout mice (O'Meara et al. 2014). Comprehensive gene and bioinformatics analysis revealed IL-13 is upstream regulator of cardiomyocyte proliferation in neonatal heart regeneration (O'Meara et al 2014). Indeed, further analysis showed that cyclin A2 and cenpa protein levels were up-regulated in the mice after IL-13 injection (new Figures EV4G, H). Thus, the fact that IL-13 secondarily triggers the upregulation of multiple different regulators could be one explanation for its high efficiency to rescue the *Gata4* mutant phenotype. Another one might be that we administered IL-13 at higher than physiological levels in a "therapeutic" approach.

We also added a cautionary note with regard to IL-13 in our discussion section (page 10, lines 28-32): *"As another note of caution, we would like to point out that we do not attribute the entirety of GATA4 dependent regenerative defects to IL-13, since other target genes of this transcription factor (such as Ccna2, Cenpa, Vegfa, Igf2r) will certainly contribute, although some of these genes (such as Cenpa) might be upregulated when IL-13 is administered in a therapeutic dose as we show in this study."*

Chromatin immunoprecipitation (ChIP) assay revealed binding of GATA4 to the *Il13* promoter in neonatal mouse hearts after cryoinjury, suggesting direct regulation of *Il13* by GATA4 (new Figure EV4F). However, we do not exclude that also indirect mechanisms might contribute to the observed correlation of IL13 and GATA4 abundance.

12. The significant differences between this study and that of Yu and co-workers (2016), regarding CM hypertrophy and the regulation of Fgf16 are not adequately explained in the discussion on page 9; especially given that the approach of cardiomyocyte-specific ko and cryo-injury were common to both studies. Why do the authors think they failed to observe a hypertrophic response or consistent down-regulation of Fgf16 in a loss of GATA4 background? Might this be explained by their constitutive CM-KO potentially having subtle effects during development, versus the inducible CM-KO targeted to neonatal stages adopted by Yu et al? How do the authors explain Yu and colleagues observations of partial rescue of regeneration by administration of AAV9-Fgf16 in light of their findings?

We addressed the differences between our study and the one by Yu et al now in much more detail in the revised manuscript in the discussion section (page 9, line 33 – page 10, line 9):” *While this manuscript was in its final stage of preparation, a different group reported that cardiac regeneration in response to transmural cryoinfarction and apical resection surgery was hampered in cardiomyocyte specific neonatal Gata4 knock-out mice, which were generated using a doxycycline responsive, troponin T dependent cardiomyocyte Cre in combination with the same Gata4 flox mouse line used in this study (Yu et al, 2016). Defective regeneration in the Gata4 mutant mice was also accompanied by reduced cardiomyocyte cell division and reduced angiogenesis, but - in contrast to our study - with markedly increased cardiomyocyte hypertrophy, which is, however, in contradiction to the previously reported prohypertrophic role of GATA4 under physiological and pathological stimulation (Oka et al, 2006). One potential explanation for the difference between the two studies with regard to hypertrophy might be the stronger initial myocardial injury (leading to a transmural scar) and the persistent strong reduction of systolic heart function in GATA4 mutant mice in the study by Yu et al, which might secondarily trigger compensatory hypertrophy, while the cardiac injury in our study was smaller and non-transmural and thus did not induce hypertrophy. Mechanistically, Yu et al. attribute the changes in their Gata4 mutant mice mainly to reduced expression of Fgf16, because administration of Fgf16 with an AAV9 vector rescued defective regeneration in their model. In contrast, we did not observe consistent downregulation of Fgf16 in our CM-G4-KO mice and GATA4 overexpression - an approach not used by the other group - even induced a strong suppression of Fgf16. Since GATA4 overexpression markedly promoted cardiomyocyte proliferation and improved myocardial regeneration, Fgf16 might not be the main downstream mediator of GATA4 during regeneration, although it clearly promotes cardiac regeneration upon overexpression as shown by Yu et al.”*

We cannot completely exclude that some subtle differences in development might be present, although complete GATA4 knock-out with the β -MHC Cre occurs only shortly before birth (E18), while Yu et al start activating their Cre even earlier (at E16.5 by administering doxycycline), thus in terms of timing the difference between both studies is not very big. In addition, the different Cre lines used might exert differential effects per se and Yu et al (in contrast to us, Figure EV1C) did not test the effects of their Cre on myocardial regeneration on a wild-type mouse background. In addition, doxycycline might exert additional effects on regeneration. We do not dispute that FGF16 induces myocardial proliferation, as it also has this effect during heart development (as shown by Lavine KJ et al., Dev Cell, 2005); we just do not find that it acts downstream of GATA4. GATA4 re-constitution in our system is highly effective in promoting regeneration, but this maneuver even suppresses FGF16, so this cannot account for the pro-regenerative effect of GATA4.

2nd Editorial Decision

11 November 2016

Thank you for the submission of your revised manuscript to EMBO Molecular Medicine. We have now heard back from the three Reviewers whom we asked to evaluate your manuscript.

I apologise for delay in getting back to you. We experienced difficulties in obtaining the reviewer evaluations in a timely manner. In addition to this, your case required further discussion with my

colleagues on the way forward.

You will see that while reviewers 1 and 2 are now satisfied that their concerns have been adequately addressed, reviewer 3 is instead remains quite reserved and points to number of important pending issues, especially his/her concerns on novelty. S/he also reiterates some technical concerns that s/he feels have not been adequately addressed.

Although we would normally not allow a second significant revision, based on the reviewer evaluations and our discussions I am prepared in this case, to give you the opportunity to improve your manuscript by responding to each point in a rebuttal and amending the manuscript where necessary. I will not be asking you to perform additional experimentation at this stage, but I do encourage you to integrate the manuscript with additional pertinent data if available. Depending on the completeness of your response, I may be able to make an editorial decision on your next, final version.

Please submit your revised manuscript within two weeks. I look forward to seeing a revised form of your manuscript as soon as possible.

I look forward to reading a new revised version of your manuscript as soon as possible.

***** Reviewer's comments *****

Referee #1 (Remarks):

well done

Referee #2 (Comments on Novelty/Model System):

The work has many detailed analyses and addresses well differened aspects for cardiac repair in early rodent life. Novelty is somewhat hampered due to previous work on the subject but I feel current works contributes to the ongoing regenerative discussion and might raise impact in the field.

Referee #2 (Remarks):

no further questions and I would like to thank the authors for a clear explanation

Referee #3 (Remarks):

In the revised manuscript from Mohammadi and colleagues the authors have attempted to address a number of the issues raised in review and the study overall is improved. However, there remains the major caveat of novelty, above and beyond the previous published study in Development by Yu et al. (2016). The novelty issue is not addressed in rebuttal and instead the authors have attempted to explain differences in their findings with that of Yu et al., (2016) by citing differences in mode/severity of injury. Most notably, they attribute a lack of a cardiomyocyte (CM) hypertrophic response in their hands (as compared to Yu et al.) to a more mild form of injury and response; however, Yu et al. also utilised cryo-injury, in addition to apical resection, so this in itself calls into question the validity of interpreting cellular responses to cryo-injury alone as reliably informative. In addition, the author's response to the query around complete rescue of both scar size and number of PH3+ CMs in cryo-injured CM-G4-KO mutants by IL-13 is somewhat unsatisfactory. They claim multiple (as yet unidentified) downstream effectors of IL-13 to explain the rescue and do not comment on roles for other potentially important factors such as Vegfa and Igf2r. The authors claim no rescue by IL-13 of the reduced neo-vascularisation/angiogenesis observed in the CM-G4-KO mutants post-cryo-injury. This suggests neovascularisation is not an important contributor to myocardial regeneration and that it does not affect scarring in the neonatal setting. This contradicts a number of previously published studies, including the fact that a robust angiogenic response is associated with heart regeneration in neonatal mice (Porrello et al., 2013) alongside the findings of Aurora et al., (2014), who demonstrated depletion of pro-angiogenic macrophages blocked

regeneration in P1 neonatal hearts post-MI. The implication of IL-13 in the current study is a novel addition as compared to the Yu et al., (2016) paper, however, the difficulty in reconciling the reported "complete rescue" detracts from this aspect.

Specific issues that remain:

Previous point 3). The authors have not adequately assessed the coronary vasculature of the CM-G4-KO mutants; they retain CD31 as a single staining of the vascular endothelium despite requests to include other markers, such as endomucin, endoglin, SM22alpha, SM-MHC etc. In addition, how have they determined capillary vessel density based on the CD31 staining shown in Fig. 2B? Point 6). The WGA stained CM data in Fig. EV2G is presumably two representative sections from the dataset comprising 3 sections per mouse, immunostained for WGA and assessed for cardiomyocyte area in 3 high-power fields (at 400x magnification) per section. As these sections are presented it is difficult to see how they can be used for an accurate assessment of CM size. Given the importance of hypertrophy (or not), in the context of the prior study by Yu and colleagues, the authors need to investigate further and present section data that lends confidence in excluding effects such as section orientation and this ought to be supported by isolation of wild type versus CM-G4-KO CMs for individual measurements.

Point 8). The refute of the use of BrdU is disappointing, given neither the half-like nor issues with administration to neonates are not relevant: a single pulse can label cells in S-phase, as representative of cell cycle activity, and intraperitoneal or subcutaneous injection of pups has been widely published elsewhere; for eg. in labelling hippocampal cells post-injury (Bartley et al., 2005) and labelling in neonatal/P1 hearts post-resection injury Porrello et al., (2011), Han et al., (2015) and indeed Yu et al., (2016) in the same directly comparable Gata4 study published in Development. PH3 is a marker of mitosis but is only indicative of cell cycle activity and not cytokinesis/hyperplasia. The bar to assess cardiomyocyte proliferation in vivo has been raised in the field, and so as a minimum it is important to utilise multiple cell cycle markers in combination: Ki67, PH3, BrdU/EdU, in addition to aurora B kinase. Yu et al. (2016) employed immunostaining for cell proliferation markers PH3, EdU and Ki67 with cardiomyocyte markers ACTN2 or TNNI3 on heart sections of Gata4 knockout or littermate controls after injury.

Point 11). The new ChIP-PCR data to indicate GATA4 directly binds IL-13 appears incomplete as it stands, in that it lacks an antibody control, for eg via the use of CM-G4-KO hearts, and moreover, there is no indication which of the numerous GATA sites in the -929 bp region of IL-13 is represented by the PCR data presented(?).

2nd Revision - authors' response

25 November 2016

Reviewer #3:

In the revised manuscript from Mohammadi and colleagues the authors have attempted to address a number of the issues raised in review and the study overall is improved. However, there remains the major caveat of novelty, above and beyond the previous published study in Development by Yu et al. (2016). The novelty issue is not addressed in rebuttal and instead the authors have attempted to explain differences in their findings with that of Yu et al., (2016) by citing differences in mode/severity of injury.

Compared to the study by Yu et al. (2016), our study introduces a number of new important findings with regard to the role of GATA4 in neonatal heart regeneration:

1) We found that endogenous myocardial GATA4 protein expression is high at postnatal day (P) 1, but sharply declines towards postnatal day 7, when regeneration is no longer possible. Therefore, we identified for the first time endogenous postnatal GATA4 regulation as one reason for loss of regenerative capacity of mice in the first postnatal week. Importantly, with our β -MHC based Cre approach, we can blunt endogenous cardiac GATA4 expression at P1 (Figure 1C in our manuscript), which is not so clear in the inducible iTNT-Cre approach by Yu et al., who only assessed GATA4 levels at P4.

2) We show for the first time, that re-introduction of GATA4 in the heart at P7 to increase the levels towards what is seen at P1 can markedly improve cardiac regeneration at P7. This finding can eventually be of high translational relevance in the future. Yu et al. did not employ any GATA4 based gain of function approach.

3) We performed extensive in vitro studies in isolated cardiomyocytes with loss of GATA4 (in fetal cardiomyocytes and in explant culture) as well as GATA4 overexpression (in neonatal cardiomyocytes), in order to evaluate the cell autonomous role of GATA4 in cardiomyocytes. We found that downregulation and overexpression of GATA4 leads to reduced or increased cardiomyocyte proliferation, respectively, indicating that even without the effects of GATA4 on endothelial cells/angiogenesis, it can directly drive cardiomyocyte proliferation without affecting cell death.

4) With IL-13, Igf2r, Tsc22d1 (among others) we identified interesting new candidate genes of GATA4 in cardiac regeneration.

5) We excluded effects of epicardial and inflammatory cells for the regenerative effects of GATA4.

6) Also with potential high translational impact for the future, we found that reduced endogenous IL-13 levels in the myocardium of the CM-G4-KO mice contribute to impaired regeneration of these mice, since exogenous administration of IL-13 can normalize the regenerative capacity. Therefore, one should test in the future, whether IL-13 administration can improve cardiac regeneration after P7, when endogenous GATA4 level are down. If successful, an IL-13 protein therapy approach could carefully be tested in larger animals, before it could eventually be developed towards the clinic.

Most notably, they attribute a lack of a cardiomyocyte (CM) hypertrophic response in their hands (as compared to Yu et al.) to a more mild form of injury and response; however, Yu et al. also utilised cryo-injury, in addition to apical resection, so this in itself calls into question the validity of interpreting cellular responses to cryo-injury alone as reliably informative.

We addressed at length in our discussion, why we performed cryoinjury instead of apical resection surgery as model for neonatal regeneration (page 8 of the revised manuscript):

“We used cryoinjury as model to study neonatal heart regeneration, because it generates highly reproducible myocardial lesions (Jesty et al, 2012; Polizzotti et al, 2016; Polizzotti et al, 2015), as opposed to cardiac apex resection or LAD ligation, which trigger lesions more variable in size. In addition, apex resection and LAD ligation - although very useful as models for basic research - might be less relevant with regard to human disease as myocardial amputation does not happen in patients and coronary artery blockage only rarely occurs in children (Polizzotti et al, 2016). While cardiac cryoinjury is certainly also not found in patients, the pathological changes it induces, such as inflammation, scar formation, and reduced or unchanged levels of cardiomyocyte mitosis and cell division were also reported in pediatric patients with heart disease (e.g. tetralogy of Fallot) (Polizzotti et al, 2015; Wald et al, 2009). Similar to what was previously shown, we did not find complete cardiac regeneration after cryoinjury, since a very small scar was still detectable 60 days later in control mice (Babu-Narayan et al, 2006; Darehzereshki et al, 2015; Jesty et al, 2012; Polizzotti et al, 2015).”

In addition our cryoinjury model is definitively more mild compared to that used by Yu et al., since we used a 0.8mm tip and only applied it to the myocardium for 3 seconds, while Yu et al. used a 1mm tip and applied it for 7-8 seconds (page 946 in their paper). When we initially established our cryoinjury model, we tried to apply the cryoprobe for 7s, but with that approach the mortality of the neonatal mice was >50% (regardless of the genotype). The increased injury in the paper by Yu et al., becomes also evident by the clear transmural injury that these authors find, while our injury was clearly non-transmural. Our hypothesis is that the increased injury leads to more marked cardiac dysfunction in the CM-G4-KO mice of Yu et al., which secondarily induces hypertrophy. This hypertrophy is also at odds with preexisting literature, because Oka et al demonstrated in 2006 that CM-G4-KO mice exert a reduced hypertrophic response (Circ Res, 2006). Our model avoids this secondary hypertrophy in the CM-G4-KO mice by inducing a smaller myocardial injury. We confirmed the lack of increased cardiac hypertrophy in our CM-G4-KO mice by analyzing

hypertrophic marker genes in our revised manuscript (revised Figure EV2H). We also address this issue in the discussion of our manuscript (page 9).

In addition, the author's response to the query around complete rescue of both scar size and number of PH3+ CMs in cryo-injured CM-G4-KO mutants by IL-13 is somewhat unsatisfactory. They claim multiple (as yet unidentified) downstream effectors of IL-13 to explain the rescue and do not comment on roles for other potentially important factors such as Vegfa and Igf2r.

It is, in fact, not uncommon in the field that “therapeutic measures” completely rescue scar size and/or cardiac function: For example, administration of the antioxidant n-acetyl cysteine rescued the scar size after apical resection towards control levels (Tao G et al., Nature 2016). In addition, several measures, such as early neuregulin-1 administration or constitutive activation of ErBB2 even rescue the cardiac function after neonatal heart injury towards sham treated animals (without any cardiac injury), see Polizzotti BD et al., 2015 Sci Transl Med and D`Uva G et al Nat Cell Biol, 2015.

In addition two main arguments make our complete rescue in CM-G4-KO mice towards levels of control mice after injury (which we now also confirmed by echocardiography in the revised manuscript, Figure 5C) plausible:

1) IL-13 was identified as a central upstream master inducer of cardiac regeneration (especially cardiomyocyte proliferation) by O`Meara CC et al., (Circ Res 2015) through comprehensive RNA-sequencing of multiple in vitro and in vivo models of cardiac regeneration and subsequent bioinformatic integration of all models. Therefore, IL-13 will induce multiple downstream targets to promote regeneration. As an example, we found that cyclinA2 and cenpa levels (two central mediators of cardiomyocyte division that are down in the hearts of our CM-G4-KO mice) are increased by this treatment to wild-type levels. The reviewer is right that more downstream targets of IL-13 need to be identified in the future, but its pro-regenerative effect in CM-G4-KO is without doubt.

2) We administered IL-13 at a higher level than what would be expected endogenously even in wild-type mice, therefore, the dose of IL-13 might have been in the therapeutic range, which might be able to overcome more regenerative barriers as compared to endogenous IL-13 and therefore was powerful enough to completely restore regeneration towards wild-type levels in CM-G4-KO mice. Both points are mentioned in the Discussion.

Page 9:” We attribute GATA4 mediated regeneration at least in part to occur via IL-13, a pleiotropic T helper-2-type cytokine, which was recently reported as important upstream inducer of mitosis in isolated neonatal rat cardiomyocytes acting through its receptor IL13Ra1 on these cells (O`Meara et al, 2015). “

Page 10:” As another note of caution, we would like to point out that we do not attribute the entirety of GATA4 dependent regenerative defects to IL-13, since other target genes of this transcription factor (such as *Ccna2*, *Cenpa*, *Vegfa*, *Igf2r*) will certainly contribute, although some of these genes (such as *Cenpa*) might be upregulated when IL-13 is administered in a therapeutic dose as we show in this study.”

We agree with this reviewer that Vegfa and Igf2r are important molecules, which require further attention to fully understand their role in cardiac regeneration.

The authors claim no rescue by IL-13 of the reduced neo-vascularisation/angiogenesis observed in the CM-G4-KO mutants post-cryo-injury. This suggests neovascularisation is not an important contributor to myocardial regeneration and that it does not affect scarring in the neonatal setting. This contradicts a number of previously published studies, including the fact that a robust angiogenic response is associated with heart regeneration in neonatal mice (Porrello et al., 2013) alongside the findings of Aurora et al., (2014), who demonstrated depletion of pro-angiogenic macrophages blocked regeneration in P1 neonatal hearts post-MI. The implication of IL-13 in the current study is a novel addition as compared to the Yu et al.,

(2016) paper, however, the difficulty in reconciling the reported "complete rescue" detracts from this aspect.

We do not at all dispute the fact that angiogenesis is important for cardiac regeneration. We state, for example that both reduced cardiomyocyte proliferation as well as decreased cardiac angiogenesis contribute to the regenerative deficiency in cardiomyocyte specific GATA4 knock-out mice.

Page 6:” Since therefore decreased cardiomyocyte proliferation and angiogenesis are the most likely reasons for impaired heart regeneration in CM-G4-KO mice, we employed...”

However, it is also likely that therapeutic manipulation to selectively enhance cardiomyocyte proliferation can also effectively improve cardiac regeneration, even without increased angiogenesis. This for example is the case in cardiomyocyte specific cyclinD2 overexpression mice, which show enhanced myocardial regeneration and myocyte proliferation even at the adult stage after myocardial infarction (Pasumarthi KBS et al., *Circ Res*, 2005). In addition, as discussed above, we supplied IL-13 in a therapeutic dose, which might have overcome regenerative blocks (as for example the lack of angiogenesis enhancement in CM-G4-KO mice).

In addition, Yu et al have not even examined angiogenesis in their FGF16 rescue attempt (Yu et al., 2016, *Development*).

Specific issues that remain:

Previous point 3). The authors have not adequately assessed the coronary vasculature of the CM-G4-KO mutants; they retain CD31 as a single staining of the vascular endothelium despite requests to include other markers, such as endomucin, endoglin, SM22alpha, SM-MHC etc. In addition, how have they determined capillary vessel density based on the CD31 staining shown in Fig. 2B?

We now address this point in our revised manuscript, where we immunostained for α -smooth-muscle-actin (α SMA) to determine the myocardial abundance of small conductance vessels (~20-50 μ m in diameter) after GATA4 over-expression as well as in CM-G4-KO mice with and without IL-13 treatment. These data is shown in the revised Figure 5F and in the revised Figure EV4F. As demonstrated there GATA4 (neither its overexpression, nor its deletion) and also not IL-13 administration had a significant effect on the abundance of small conductance vessels in the neonatal myocardium after injury.

We quantified CD31 positive capillaries by counting their abundance in high power fields only in the green channel (showing CD31, as depicted on the right in Figure 2B) and subsequently we counted the number of cardiomyocytes in the red channel showing WGA encircled cells. We have published this approach before (Heineke et al., 2007, *JCI* and Appari M et al., *Circ Res*, 2016).

Point 6). The WGA stained CM data in Fig. EV2G is presumably two representative sections from the dataset comprising 3 sections per mouse, immunostained for WGA and assessed for cardiomyocyte area in 3 high-power fields (at 400x magnification) per section. As these sections are presented it is difficult to see how they can be used for an accurate assessment of CM size. Given the importance of hypertrophy (or not), in the context of the prior study by Yu and colleagues, the authors need to investigate further and present section data that lends confidence in excluding effects such as section orientation and this ought to be supported by isolation of wild type versus CM-G4-KO CMs for individual measurements.

Determination of cardiomyocyte cross-sectional area is a widely accepted measure of cardiac hypertrophy (see for example Boström P et al., *Cell*, 2010; Anand P et al., *Cell*, 2013; Lee DI et al., *Nature* 2015; all these papers show pictures similar to what we demonstrate in Figure EV2G). Also Yu et al. use this measure similar as we apply it (Figure 2F in their manuscript on page 938 in *Development*, 2016). However, we now also measured hypertrophic marker gene expression in the revised manuscript by qPCR. As demonstrated in the revised Figure EV2H, we do not see enhanced Nppa or Nppb expression in CM-G4-KO mice after cryoinjury or sham surgery compared to control mice (there is even a trend towards reduced levels in CM-G4-KO in accordance with the literature, see Oka et al., *Circ Res* 2006).

In addition, we have demonstrated how we measured the cardiomyocyte cross sectional area in the particular section presented in the revised Figure EV2H.

Point 8). The refute of the use of BrdU is disappointing, given neither the half-like nor issues with administration to neonates are not relevant: a single pulse can label cells in S-phase, as representative of cell cycle activity, and intraperitoneal or subcutaneous injection of pups has been widely published elsewhere; for eg. in labelling hippocampal cells post-injury (Bartley et al.,2005) and labelling in neonatal/P1 hearts post-resection injury Porrello et al., (2011), Han et al., (2015) and indeed Yu et al., (2016) in the same directly comparable Gata4 study published in Development. PH3 is a marker of mitosis but is only indicative of cell cycle activity and not cytokinesis/hyperplasia. The bar to assess cardiomyocyte proliferation in vivo has been raised in the field, and so as a minimum it is important to utilise multiple cell cycle markers I combination: Ki67, PH3, BrdU/EdU, in addition to aurora B kinase. Yu et al. (2016) employed Immunostaining for cell proliferation markers PH3, EdU and Ki67 with cardiomyocyte markers ACTN2 or TNNI3 on heart sections of Gata4 knockout or littermate controls after injury.

We agree with this reviewer that we probably could have used repeated injections of BrdU to measure cell cycle activity. However as we use minipumps infusion of BrdU (now EdU) in adult mice in collaboration with Loren Field (a well-regarded expert in the field from Indianapolis, USA), and he emphasized to us that this is the best way to obtain consistent results, we did not employ this approach in our neonates, which are too small to take minipumps. We did, however, use BrdU in neonatal cardiomyocytes (Figure 3C in our manuscript).

To introduce another marker of cell cycle activity, we now stained for Ki67 in myocardial sections with/without GATA4 over-expression and also in mice with or without cardiomyocyte GATA4 deletion and IL-13 therapy 7 days after cryoinjury. As depicted in the revised Figures 4G and 5D, GATA4 overexpression/deletion also increased/decreased cardiomyocyte Ki67 labeling, while IL-13 increased Ki67 labelling in CM-G4-KO mice.

It should be noted, however, that some very prominent papers in the field only rely on two methods to demonstrate cardiomyocyte cell cycle activity (for example pH3 and aurora B):

- Mahmoud AI, Dev Cell, 2015 (pH3 and AuroraB kinase)
- Polizzotti BD, Science Translat Med, 2015 (pH3 and AuroraB kinase)
- Tao G, Nature, 2016 (only EdU)

Point 11). The new ChIP-PCR data to indicate GATA4 directly binds IL-13 appears incomplete as it stands, in that it lacks an antibody control, for eg via the use of CM-G4-KO hearts, and moreover, there is no indication which of the numerous GATA sites in the -929 bp region of IL-13 is represented by the PCR data presented(?).

In the revised manuscript, we include another primer pair ("primer2") in addition to primer1, which was previously used by Pai et al., 2008, J Immunol. With both primer pairs, we obtained around 10-fold enrichment of the Il13 promoter DNA after immunoprecipitation with anti-GATA4 from wild-type mouse hearts 7 days after cryoinjury (revised Figure EV4G). We also indicate the localization of the primers in this figure in the revised manuscript.

The GATA4 antibody has been used for ChIP before by us (Heineke et al., 2007, JCI) and was, in fact also used by Yu et al., 2016, Development. The use of IgG is the usual control for a ChIP assay and we have not tried to conduct this experiment from CM-G4-KO mice (and this was also not done by Yu et al., 2016).

Corresponding Author Name: Joerg Heineke

Journal Submitted to: Embo Molecular Medicine

Manuscript Number: EMM-2016-06602